# What and How does In-Context Learning Learn? Bayesian Model Averaging, Parameterization, and Generalization

## Abstract

In-Context Learning (ICL) ability has been found efficient across a wide range of applications, where the Large Language Models (LLM) learn to complete the tasks from the examples in the prompt without tuning the parameters. In this work, we conduct a comprehensive study to understand ICL from a statistical perspective. First, we show that the perfectly pretrained LLMs perform Bayesian Model Averaging (BMA) for ICL under a dynamic model of examples in the prompt. The average error analysis for ICL is then built for the perfectly pretrained LLMs with the analysis of BMA. Second, we demonstrate how the attention structure boosts the BMA implementation. With sufficient examples in the prompt, attention is proven to perform BMA under the Gaussian linear ICL model, which also motivates the explicit construction of the hidden concepts from the attention heads values. Finally, we analyze the pretraining behavior of LLMs. The pretraining error is decomposed as the generalization error and the approximation error, which are bounded separately. Then the ICL average error of the pretrained LLMs is shown to be the sum of $O(T^{-1})$ and the pretraining error. In addition, we analyze the ICL performance of the pretrained LLMs with misspecified examples. The theoretical findings are corroborated with the experimental results.

## 1 Introduction

With the ever-increasing sizes of model capacity and corpus, Large Language Models (LLM) have achieved tremendous successes across a wide range of tasks [Dong et al., 2019; Wei et al., 2022c; Kojima et al., 2022; Ouyang et al., 2022]. Recent studies have revealed that these LLMs possess immense potential, as their large capacity allows for a series of *emergent abilities* [Wei et al., 2022b; Liu et al., 2023].

One such ability is ICL, which enables an LLM to learn from just a few examples, without changing the network parameters. Despite the tremendous empirical successes, theoretical understanding of ICL remains limited. Specifically, existing works fail to explain why LLMs have the ability for ICL, how the attention mechanism is related to the ICL ability, and how pretraining influences ICL. Although the optimality of ICL is investigated in Xie et al. [2021] and Wies et al. [2023], these works both make unrealistic assumptions on the pretrained models, and their results cannot demystify the particular role played by the attention mechanism in ICL.

In this work, we focus on the scenario where a transformer is first pretrained on a large dataset and then prompted to perform ICL. Our goal is to rigorously understand why the practice of "pretraining + prompting" unleashes the power of ICL. To this end, we

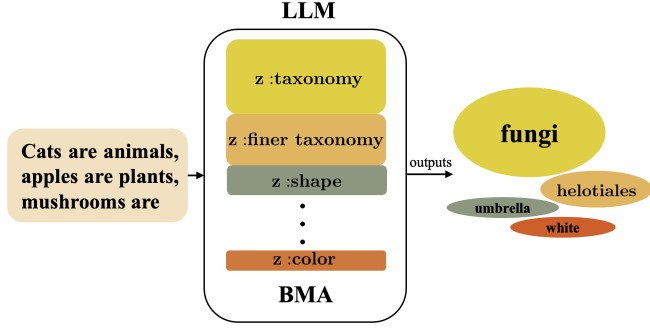

Figure 1: LLMs implement Bayesian Model Averaging (BMA) for In-Context Learning (ICL). They estimate the posterior of hidden concept $z$ from examples and use the posterior to mix the conditional probability of response on the query and hidden concept $z$.

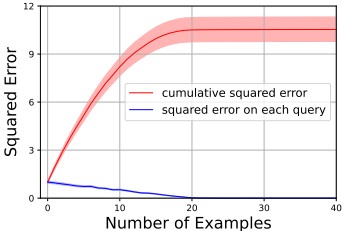
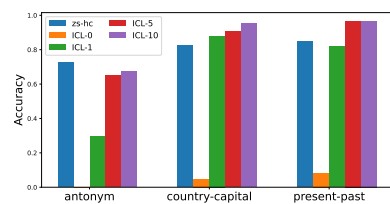
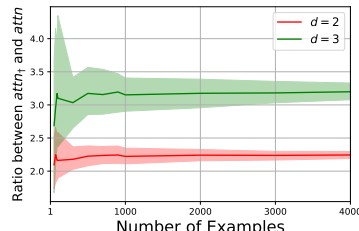

Figure 2: The cumulative squared error of LLMs trained for linear regression is bounded by a constant. It verifies Propostion 3.4, which is a result of Propostion 3.3.

Figure 3: The constructed hidden concept provides sufficient information for LLM. Conditioned on it in a zero-shot setting, LLMs have comparable performance with ICL with several examples. It verifies Propostions 3.3 and 3.6.

Figure 4: The ratios between `attn`† and `attn` converge to a constant that depends on the dimension when the example number $T$ tends to infinity. It verifies Propostion 3.6.

aim to answer the following three questions: **(a)** What type of ICL estimator is learned by LLMs? **(b)** What are suitable performance metrics to evaluate ICL accurately and what are the error rates? **(c)** What is the role played by the transformer architecture during the pretraining and prompting stages? The first and the third questions demand scrutinizing the transformer architecture to understand how ICL happens during transformer prompting. The second question then requires statistically analyzing the extracted ICL process. Moreover, the third question necessitates a holistic understanding beyond prompting — we also need to characterize the statistical error of *pretraining* and how this error affects prompting.

To address these questions, we adopt a Bayesian view and assume that the examples fed into a pretrained LLM are sampled from a hidden variable model parameterized by a hidden concept $z_* \in \mathfrak{Z}$. Moreover, the pretraining dataset contains sequences of examples from the same hidden variable model, but with the concept parameter $z \in \mathfrak{Z}$ itself randomly distributed according to a prior distribution. We mathematically formulate ICL as the problem of predicting the response of the current covariate, where the prompt contains $t$ examples of covariate-response pairs and the current covariate.

Under such a setting, to answer **(a)**, we show that the perfectly pretrained LLMs perform ICL in the form of BMA under a dynamical data model. That is, LLM first computes a posterior distribution of $z_* \in \mathfrak{Z}$ given the first $t$ examples, and then predicts the response of the $(t+1)$-th covariate by aggregating over the posterior (Proposition 3.3), which is empirically verified in Section 3.2.

In addition, to answer **(b)**, we adopt the online learning framework and define a notion called ICL average error, which is the averaged prediction error of ICL on a sequence of covariate-response examples. We prove that the ICL average error after prompting $t$ examples is $\mathcal{O}(1/t)$ up to the statistical error of the pretrained model (Theorem 5.2), which is validated by experiments in Section 3.2.

Finally, to answer **(c)**, we elucidate the role played by the transformer architecture in prompting and pretraining respectively. In particular, we show that a variant of attention mechanism encodes BMA in its architecture under a linear Gaussian model, which enables the transformer to perform ICL via prompting. Such an attention mechanism can be viewed as an extension of linear attention and coincides with the standard softmax attention [Garnelo & Czarnecki, 2023] when the length of the prompt goes to infinity. Thus we show that softmax attention Vaswani et al. [2017] approximately encodes BMA (Proposition 3.6), which is empirically verified in Section 3.4. Besides, the transformer architecture enables a fine-grained analysis of the statistical error incurred by pretraining. In particular, we prove that the error of the pretrained language model, measured via total variation, is bounded by a sum of approximation error and generalization error (Theorem 4.3). The approximation error decays to zero exponentially fast as the depth of the transformer increases (Proposition 4.4), while the generalization error decays to zero sublinearly with the number of tokens in the pretraining dataset. This features the first pretraining analysis of transformers in total variation distance that also takes the approximation error into account. Furthermore, as an interesting extension, we also study the misspecified case where the response variables of the examples fed into the LLM are perturbed.

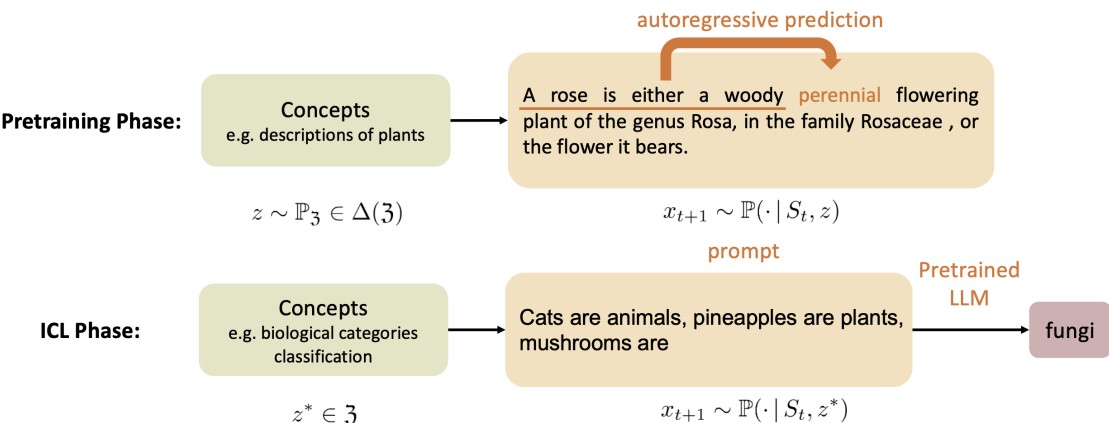

Figure 5: To form the pretraining dataset, a hidden concept $z$ is first sampled according to $\mathbb{P}_{\mathfrak{Z}}$, and a document is generated from the concept. Taking the token sequence $S_t$ up to position $t \in [T]$ as the input, the LLM is pretrained to maximize the next token $x_{t+1}$. During the ICL phase, the pretrained LLM is prompted with several examples to predict the response of the query.

We provide sufficient conditions for ICL to be robust to the perturbations and establish the finite-sample statistical error (Proposition G.5).

In sum, by addressing questions (a)–(c), we provide a unified understanding of the ICL ability of LLMs and the particular role played by the attention mechanism. Our theory provides a holistic theoretical understanding of the ICL average error, approximation, and generalization errors of ICL.

## 2 Preliminary

**Attention and Transformers.** Attention mechanism has been the most powerful and popular neural network module in both Computer Vision (CV) and Natural Language Processing (NLP) communities, and it is the backbone of the LLMs [Devlin et al., 2018; Brown et al., 2020]. Assume that we have a query vector $q \in \mathbb{R}^{d_k}$. With $T$ key vectors in $K \in \mathbb{R}^{T \times d_k}$ and $T$ value vectors in $V \in \mathbb{R}^{T \times d_v}$, the attention mechanism maps the query vector $q$ to $\mathtt{attn}(q, K, V) = V^\top \mathtt{softmax}(Kq)$, where $\mathtt{softmax}$ normalizes a vector via the exponential function, i.e., for $x \in \mathbb{R}^d$, $[\mathtt{softmax}(x)]_i = \exp(x_i) / \sum_{j=1}^d \exp(x_j)$ for $i \in [d]$. The output is a weighted sum of $V$, and the weights reflect the closeness between $W$ and $q$. For $t$ query vectors, we stack them into $Q \in \mathbb{R}^{t \times d_k}$. Attention maps these queries using the function $\mathtt{attn}(Q, K, V) = \mathtt{softmax}(QK^\top)V \in \mathbb{R}^{t \times d_v}$, where $\mathtt{softmax}$ is applied row-wisely. In the practical design of transformers, practitioners usually use Multi-Head Attention (MHA) instead of single attention to express sophisticated functions, which forwards the inputs through $h$ attention modules in parallel and outputs the sum of these sub-modules. Here $h \in \mathbb{N}$ is a hyperparameter. Taking $X \in \mathbb{R}^{T \times d}$ as the input, MHA outputs $\mathtt{mha}(X, W) = \sum_{i=1}^h \mathtt{attn}(XW_i^Q, XW_i^K, XW_i^V)$, where $W = (W_i^Q, W_i^K, W_i^V)_{i=1}^h$ is the parameters set of $h$ attention modules, $W_i^Q \in \mathbb{R}^{d \times d_h}$, $W_i^K \in \mathbb{R}^{d \times d_h}$, and $W_i^V \in \mathbb{R}^{d \times d}$ for $i \in [h]$ are weight matrices for queries, keys, and values, and $d_h$ is usually set to be $d/h$ [Michel et al., 2019]. The transformer is the concatenation of the attention modules and the fully-connected layers, which is widely adopted in LLMs [Brown et al., 2020].

**Large Language Models and In-Context Learning.** Many LLMs are *autoregressive*, such as GPT [Brown et al., 2020]. It means that the model continuously predicts future tokens based on its own previous values. For example, starting from a token $x_1 \in \mathfrak{X}$, where $\mathfrak{X}$ is the alphabet of tokens, a LLM $\mathbb{P}_\theta$ with parameter $\theta \in \Theta$ continuously predicts the next token according to $x_{t+1} \sim \mathbb{P}_\theta(\cdot \mid S_t)$ based on the past $S_t = (x_1, \cdots, x_t)$ for $t \in \mathbb{N}$. Here, each token represents a word and the position of the word [Ke et al., 2020], and the token sequences $S_t$ for $t \in \mathbb{N}$ live in the sequences space $\mathfrak{X}^*$. LLMs are first *pretrained* on a huge body of corpus, making the prediction $x_{t+1} \sim \mathbb{P}_\theta(\cdot \mid S_t)$ accurate, and then prompted to perform downstream tasks. During the pretraining phase, we aim to maximize the conditional probability $\mathbb{P}_\theta(x \mid S)$ over the nominal next token $x$ [Brown et al., 2020].

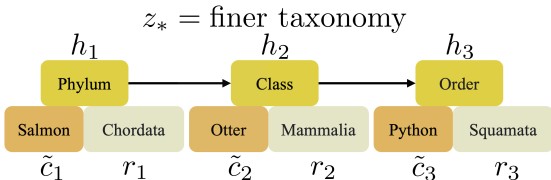

Figure 6: The hidden concept $z_*$ is "finer taxonomy" makes the hidden variables $h_t$ evolve to finer level according to equation 3.1. These hidden variables parameterize the relationship between $\widetilde{c}_t$ and $r_t$ with equation 3.2.

After pretraining, LLMs are prompted to perform downstream tasks without tuning parameters. Different from the finetuned models that learn the task explicitly [Liu et al., 2023], LLMs can implicitly learn from the examples in the *prompt*, which is known as ICL [Brown et al., 2020]. Concretely, pretrained LLMs are provided with a prompt $\mathtt{pt}_t = (\widetilde{c}_1, r_1, \ldots, \widetilde{c}_t, r_t, \widetilde{c}_{t+1})$ with $t$ examples and a query as inputs, where each pair $(\widetilde{c}_i, r_i) \in \mathfrak{X}^* \times \mathfrak{X}$ is an example of the task, and $\widetilde{c}_{t+1}$ is the query, as shown in Figure 5 in Appendix **??**. For example, the $\mathtt{pt}_t$ with $t = 2$ can be "Cats are animals, apples are plants, mushrooms are". Here $\widetilde{c}_1 \in \mathfrak{X}^*$ is a token sequence "Cats are", while $r_1$ is the response "animals". The query $\widetilde{c}_{t+1}$ is "mushrooms are", and the desired response is "fungi". The prompts are generated from a hidden concept $z_* \in \mathfrak{Z}$, e.g., $z_*$ can be the classification of biological categories, where $\mathfrak{Z}$ is the concept space. The generation process is $\widetilde{c}_i \sim \mathbb{P}_{\mathrm{q}}$ and $r_i \sim \mathbb{P}(\cdot \,|\, \mathtt{pt}_{i-1}, z_*)$ for the nominal distribution $\mathbb{P}$ and $i \in [t]$, where $\mathbb{P}_{\mathrm{q}}$ is the covariate distribution. Thus, when performing ICL, LLMs aim to estimate the conditional distribution $\mathbb{P}(r_{t+1} | \mathtt{pt}_t, z_*)$. It is widely conjectured and experimentally found that the pretrained LLMs can implicitly identify the hidden concept $z_* \in \mathfrak{Z}$ from the examples, and then perform ICL by outputting from $\mathbb{P}(r_{t+1} | \mathtt{pt}_t, z_*)$. In the following, we will provide theoretical justifications for this claim. We note that delimiters are omitted in our work, and our results can be generalized to handle this case. Since LLMs are autoregressive, the definition of the notation $\mathbb{P}(\cdot \,|\, S)$ with $S \in \mathfrak{X}^*$ may be ambiguous because the length of the subsequent tokens is not specified. Unless explicitly specified, we let $\mathbb{P}(\cdot \,|\, S)$ denote the distribution of the next single token conditioned on $S$.

## 3 In-Context Learning via Bayesian Model Averaging

In this section, we show that LLMs perform ICL implicitly via BMA from two perspectives. First, we show that the statistical model of ICL and the training loss enable LLMs to implement BMA for ICL under a dynamical data model. Second, we show that the self-attention mechanism boosts the implementation of the BMA algorithm in LLMs.

### 3.1 ICL Statistical Models Enables Bayesian Model Averaging

Given a sequence $S = \{(\widetilde{c}_t, r_t)\}_{t=1}^T$ with $T$ examples generated from a hidden concept $z_* \in \mathfrak{Z}$, we use $S_t = \{(\widetilde{c}_i, r_i)\}_{i=1}^t$ to represent the first $t$ ICL examples in the sequence. Here $\widetilde{c}_t$ and $r_t$ respectively denote the ICL covariate and response. During the ICL phase, a LLM is sequentially prompted with $\mathtt{pt}_t = (S_t, \widetilde{c}_{t+1})$ for $t \in [T-1]$, i.e., the first $t$ examples and the $(t+1)$-th covariate. The prompted LLM aims to predict the response $r_{t+1}$ based on $\mathtt{pt}_t = (S_t, \widetilde{c}_{t+1})$ whose true distribution is $r_{t+1} \sim \mathbb{P}(\cdot \,|\, \mathtt{pt}_t, z_*)$. For the analysis of ICL, we focus on the following hidden variable model

**Assumption 3.1** (Dynamic Hidden Variable Model). The hidden concept $z_* \in \mathfrak{Z}$ parameterizes the distributions of hidden variables $\{h_t\}_{t=1}^T \in \mathcal{H}^T$ as

$$h_t = g_{z_*}(h_1, \ldots, h_{t-1}, \zeta_t), \tag{3.1}$$

where $g_{z_*}$ is a function parameterized by $z_*$, and $\{\zeta_t\}_{t=1}^T$ are exogenous noises. These hidden variables parameterize the covariates and responses for ICL as

$$r_t = f(\widetilde{c}_t, h_t, \xi_t), \quad \widetilde{c}_t \sim \mathbb{P}_{\mathrm{q}} \quad \forall t \in [T], \tag{3.2}$$

where $\mathbb{P}_q$ is the distribution of query, the hidden variable $h_t \in \mathcal{H}$ determines the relation between $c_t$ and $r_t$, $\xi_t \in \Xi$ for $t \in [T]$ are i.i.d. random noises, and $f : \mathcal{X}^* \times \mathcal{H} \times \Xi \to \mathfrak{X}$ is a function that relates response $r_t$ to $\widetilde{c}_t, h_t$, and $\xi_t$.

In the data generation process, a hidden concept $z_* \in \mathfrak{Z}$ is first generated from $\mathbb{P}(z)$. The covariates $\widetilde{c}_t$ are generated from $\mathbb{P}_q$ in an i.i.d. manner. Then the hidden variables $\{h_t\}_{t=1}^T$ and responses $\{r_t\}_{t=1}^T$ are generated according to equation 3.1 and equation 3.2. The model in Assumption 3.1 essentially assumes that the hidden concept $z_*$ implicitly determines the transition of the conditional distribution $\mathbb{P}(r_t = \cdot \,|\, \widetilde{c}_t)$ by affecting the evolution of the hidden variables $\{h_t\}_{t\in[T]}$. These hidden variables capture the relationship between examples $\{(\widetilde{c}_t, r_t)\}_{t=1}^T$, since the examples may have some inferent relationship and share some similarity when they are come up with by humans [Elman, 1995; Niyogi et al., 1997]. In other words, this assumption assumes that the examples convey a *main semantic* $z_*$, e.g., "finer taxonomy" in Figure. 6. Then the response is generated from the hidden variable $h$, e.g., "class", and covariate, e.g., "otter" in Figure. 6. This model is quite general, and it subsumes the models in previous works.

**Comparision with existing models** The models in the existing works all assume that the examples are i.i.d., i.e., $h_t = g_{z_*}(\zeta_t)$ in the model of Assumption 3.1. For example, the Hiddn Markov Model (HMM) model in Xie et al. [2021] assumes that the hidden variables for each example are independently generated. In contrast, we allow them to depend on each other via equation 3.1. When the hidden variables $h_t = z_*$ for $t \in [T]$ degenerate to the hidden concept, the model in Assumption 3.1 recovers the topic model in Wang et al. [2023] and the ICL model in Jiang [2023].

Assuming that the tokens follow the statistical model given in equation 3.2, during pretraining, we collect $N_p$ independent trajectories by sampling from equation 3.2 with concept $z$ randomly sampled from $\mathbb{P}(z)$. Intuitively, during pretraining, by training in an autoregressive manner, the LLM approximates the conditional distribution $\mathbb{P}(r_{t+1} \,|\, \mathtt{pt}_t) = \mathbb{E}_{z \sim \mathbb{P}(z \,|\, \mathtt{pt}_t)}[\mathbb{P}(r_{t+1} \,|\, \mathtt{pt}_t, z)]$, which is the conditional distribution of $r_{t+1}$ given $\mathtt{pt}_t$, aggregated over the randomness of the concept $z_*$. Given an infinite number of samples and the sufficiently large function class $\Theta$, the pretrained LLMs can perfectly match the pretraining distribution.

**Assumption 3.2** (Perfect Pretraining). A LLM $\mathbb{P}_\theta$ is called perfectly pretrained if for all $\mathtt{pt} \in \mathcal{X}^*$ and $r \in \mathcal{X}$, we have $\mathbb{P}_\theta(r \,|\, \mathtt{pt}) = \mathbb{P}(r \,|\, \mathtt{pt})$, where $\mathbb{P}$ is the distribution induced by the model in Assumption 3.1. We will relax this assumption in Section 4 by analyzing the pretraining error.

**Proposition 3.3** (LLMs Perform BMA). Under Assumptions 3.1 and 3.2, LLMs perform BMA for ICL, i.e.,

$$\mathbb{P}_\theta(r_{t+1} \,|\, \mathtt{pt}_t) = \int \mathbb{P}(r_{t+1} \,|\, \widetilde{c}_{t+1}, S_t, z) \mathbb{P}(z \,|\, S_t) \mathrm{d}z, \tag{3.3}$$

where $\mathbb{P}$ is the distribution induced by Assumption 3.1.

We note that the left-hand side of equation 3.3 is the prediction of the pretrained LLM given a prompt $\mathtt{pt}_t$. Meanwhile, the right-hand side is exactly the prediction given by the BMA algorithm that infers the posterior belief of the concept $z_*$ based on $S_t$ and predicts $r_{t+1}$ by aggregating the likelihood in equation 3.2 with respect to the posterior $\mathbb{P}(z_* = \cdot \,|\, S_t)$, as shown in Figure 1. Thus, this proposition shows that perfectly pretrained LLMs are able to perform ICL because they **implement BMA during prompting**. As mentioned, Proposition 3.3 is proved under a more general model than the previous works and thus serves as a generalized result of some claims in the previous works. We note that the claim of Proposition 3.3 is independent of the network structure. This partially explains why LSTMs demonstrate ICL ability in Xie et al. [2021]. In Section 3.3, we will demonstrate how the attention mechanism helps to implement BMA. The proof of Proposition 3.3 is in Appendix E.2.

Next, we study the performance of ICL from an online learning perspective. Recall that LLMs are continuously prompted with $S_t$ and aim to predict the $(t+1)$-th covariate $r_{t+1}$ for $t \in [T-1]$. This can be viewed as an online learning problem. For any algorithm that generates a sequence of density estimators $\{\widehat{\mathbb{P}}(r_t)\}_{t=1}^T$ for predicting $\{r_t\}_{t\in[T]}$, we consider the following ICL average error as its performance metric:

$$\mathtt{AE}_t = \frac{1}{t} \sup_z \sum_{i=1}^t \left( \log \mathbb{P}(r_i \,|\, \mathtt{pt}_{i-1}, z) - \log \widehat{\mathbb{P}}(r_i) \right). \tag{3.4}$$

This ICL average error measures the performance of the estimator $\widehat{\mathbb{P}}$ compared with the best hidden concept in hindsight. For the perfectly trained LLMs, the estimator is exactly $\widehat{\mathbb{P}}(r_t) = \mathbb{P}(r_{t+1} \mid \mathtt{pt}_t)$. By building the equivalence of pretrained LLM and BMA, we have the following proposition, which shows that predicting $\{r_t\}_{t\in[T]}$ by iteratively prompting the LLM incurs a $\mathcal{O}(1/T)$ average error.

**Proposition 3.4** (ICL Average Error of Perfectly Pretrained Model). Under Assumptions 3.1 and 3.2, we have for any $t \in [T]$ that

$$\frac{1}{t}\sum_{i=1}^{t}\log\mathbb{P}_\theta(r_i \mid \mathtt{pt}_{i-1}) \geq \sup_{z\in\mathcal{Z}}\Big(\frac{1}{t}\sum_{i=1}^{t}\log\mathbb{P}(r_i \mid \mathtt{pt}_{i-1}, z) + \frac{\log\mathbb{P}_\mathcal{Z}(z)}{t}\Big).$$

Here $\mathbb{P}_\mathcal{Z}$ is the prior of the hidden concept $z \in \mathcal{Z}$. When the hidden concept space $\mathfrak{Z}$ is finite and the prior $\mathbb{P}_\mathcal{Z}(z)$ is the uniform distribution on $\mathfrak{Z}$, we have that $\mathtt{AE}_t \leq \log|\mathfrak{Z}|/t$. When the nominal concept $z_*$ satisfies that $\sup_z \sum_{i=1}^{t}\log\mathbb{P}(r_i \mid z, \mathtt{pt}_{i-1}) = \sum_{i=1}^{t}\log\mathbb{P}(r_i \mid z_*, \mathtt{pt}_{i-1})$ for any $t \in [T]$, the average error is bounded as $\mathtt{AE}_t \leq \log(1/\mathbb{P}_\mathcal{Z}(z_*))/t$.

This theorem states that the ICL average error of the perfectly pretrained model is bounded by $\log(1/\mathbb{P}_\mathcal{Z}(z_*))/t$. This is intuitive since the average error is relatively large if the concept $z_*$ rarely appears according to the prior distribution. This proposition shows that, when given sufficiently many examples, predicting $\{r_t\}_{t\in[T]}$ via ICL is almost as good as the oracle method which knows true concept $z_*$ and the likelihood function $\mathbb{P}(r_i \mid \mathtt{pt}_{i-1}, z_*)$. We state the result for the case where $\mathfrak{Z}$ is finite, and these results can be generalized to uncountable $\mathfrak{Z}$ with continuity assumptions. The proof of Proposition 3.4 and the extension is in Appendix E.3. In Section 4, we characterize the deviation between the learned model and the underlying true model.

### 3.2 Experimental validations for Propositions 3.3 and 3.4

To verify Propostion 3.3, we empirically construct the hidden concept and condition on it for inference. We construct the hidden concept vector as the average sum over prompts of the values of twenty selected attention heads, i.e., we compress the hidden concept into a vector with dimension 4096. To demonstrate the effectiveness of the constructed hidden concepts, we add these hidden concept vectors at a layer of LLMs when the model resolves the prompt with zero-shot. In Figure. 3, "zs-hc" refers to the results of LLMs that infers with learned hidden concept vectors and zero-shot prompt, and "ICL-$i$" refers to the results of LLMs prompted with $i$ examples. We consider the tasks of finding antonyms, finding the capitals of countries, and finding the past tense of words, i.e., $h_t = z_*$ in Proposition 3.3. The results indicate that the LLMs conditioned on the learned hidden concept vectors $P_\theta(r_{t+1}|\widetilde{c}_{t+1}, z_*)$ have comparable performance with the LLMs prompted with several examples $P_\theta(r_{t+1}|\mathtt{pt}_t)$. This indicates that the learned hidden concept vectors are indeed efficient compression of the hidden concepts, which proves that LLMs deduce hidden concepts for ICL and corroborates with Proposition 3.3.

To verify Proposition 3.4, we will empirically show that $t \times \mathtt{AE}_t$, i.e., the cumulative error, is upper bounded by a constant. LLMs is trained for the linear regression task from scratch, which is a representative setting studied in Garg et al. [2022]; Akyürek et al. [2022]. The examples in the prompt are $\{(x_i, y_i)\}_{i=1}^{N}$, where $x_i \in \mathbb{R}^d$, $d = 20$ and $y_i = w^T x_i$ for some $w$ sampled from Gaussian distribution. Given the Gaussian model, we adopt the squared error to approximate the logarithm of the probability. Then $t \times \mathtt{AE}_t$ of the LLMs can be well approximated by the sum of the squared error till time $t$. We note that there is pretraining error in our experiments, which can be bounded by Theorems 4.3 and 5.2. The results in Figure 2 strongly corroborate our theoretical findings. First, the results verify our claim in Proposition 3.4 that $t \cdot \mathtt{AE}_t$ can be upper bounded by a constant. Second, the line of squared error indicates that the ICL of LLMs only has a significant error when $T \leq d$, i.e., the cumulative error only increases in this region. Thus, the cumulative error of the ICL by LLMs is at most linear in $O(d/T)$. From the view of our theoretical result, discretizing the set $\{z \in \mathbb{R}^d \mid \|z\|_2 \leq d\}$ with approximation error $\delta > 0$ will result in a set with $(C/\delta)^d$ elements, where $C > 0$ is an absolute constant. Proposition 3.4 implies that the cumulative error is the sum of the $\log|\mathfrak{Z}|/T = d\log(C/\delta)/T$ and the pretraining error, which matches the simulation results. The experiment details can be found in Appendix D. We also calculate the cumulative log-likelihoods in Proposition 3.4 on time series data in C, which have constant-like upper bounds.

### 3.3 Attention Parameterizes Bayesian Model Averaging

In the following, we explore the role played by the attention mechanism in ICL. To simplify the presentation, we consider the case where the covariate $\widetilde{c}_t \in \mathfrak{X}^*$ is a single token $c_t \in \mathfrak{X}$ in this subsection. During the ICL phase, pretrained LLMs are prompted with $\mathtt{pt}_t = (S_t, c_{t+1})$ and tasked with predicting the $(t+1)$-th response $r_{t+1}$. The transformers first separately map the covariates $\widetilde{c}_i$ and responses $r_i$ for $i \in [t]$ to the corresponding feature spaces, which are usually realized by the fully connected layers. We denote these two learnable mappings as $k : \mathbb{R}^d \to \mathbb{R}^{d_k}$ and $v : \mathbb{R}^d \to \mathbb{R}^{d_v}$. Their nominal values are denoted as $k_*$ and $v_*$, respectively. The pretraining of the transformer essentially learns the nominal mappings $v_*$ and $k_*$ with sufficiently many data points. After these transformations, the attention module will take $v_i = v_*(r_i)$ and $k_i = k_*(c_i)$ for $i \in [t]$ as the value and key vectors to predict the result for the query $q_{t+1} = k_{t+1} = k_*(c_{t+1})$. To elucidate the role played by attention, we consider a Gaussian linear simplification of equation 3.2.

**Assumption 3.5** (Gaussian linear ICL model)**.** For the features $v_t = v_*(r_t)$ and $k_t = k_*(c_t)$ for $t \in [t]$, we have

$$v_t = z_* \phi(k_t) + \xi_t, \quad \forall t \in [T], \tag{3.5}$$

where $\phi : \mathbb{R}^{d_k} \to \mathbb{R}^{d_\phi}$ refers to the feature mapping in some Reproducing Kernel Hilbert Space (RKHS), $z_* \in \mathbb{R}^{d_v \times d_\phi}$ corresponds to the hidden concept, and $\xi_t \sim N(0, \sigma^2 I), t \in [T]$ are i.i.d. Gaussian noises with covariance $\sigma^2 I$. The prior of $z_*$ is $\mathbb{P}(z)$ is a Gaussian distribution $N(0, \lambda I)$.

The function in equation 3.5 is general. The generality comes from: (i) the feature mapping $\phi$ and the corresponding RKHS make Kernel Mean Embedding (KME) have sufficient expressiveness since all the operations on distribution can be captured by some operations in KME space. (ii) Two learnable mappings $v_*(\cdot)$ and $k_*(\cdot)$ are neural networks and are general enough to represent continuous functions. To specify the relationship between Assumptions 3.1 and 3.5, note that equation 3.5 can be written as

$$r_t = v_*^{-1}\Big(z_* \phi\big(k_*(c_t)\big) + \xi_t\Big) \tag{3.6}$$

if $v_*^{-1}$ is reversible, which is a realization of equation 3.2 with $h_t = z$, $\xi_t = \epsilon_t$, and $f(c, h, \xi) = v_*^{-1}(h\phi(k_*(c)) + \xi)$. In other words, equation 3.5, or equivalently equation 3.6, specifies a specialization of equation 3.2 where in the feature space, the hidden concept $z_*$ represents a transformation between the value $v$ and the key $k$. Here, we simply take this as the transformation by a matrix, which can be easily generalized by building a bijection between concepts $z$ and complex transformations. In the following, to simplify the notation, let $\mathfrak{K} : \mathbb{R}^{d_k} \times \mathbb{R}^{d_k} \to \mathbb{R}$. denote the kernel function of the RKHS induced by $\phi$. The stacks of the values and keys are denoted as $K_t = (k_1, \ldots, k_t)^\top \in \mathbb{R}^{t \times d_k}$ and $V_t = (v_1, \ldots, v_t)^\top \in \mathbb{R}^{t \times d_v}$, respectively. Consequently, the model in equation 3.5 implies that

$$\mathbb{P}(v_{t+1} \,|\, \mathtt{pt}_t) = \int \mathbb{P}(v_{t+1} \,|\, z, q_{t+1}) \mathbb{P}(z \,|\, S_t) \mathrm{d}z \propto \exp\Big(-\big\|v_{t+1} - \bar{z}_t \phi(q_{t+1})\big\|_{\Sigma_t^{-1}}^2 / 2\Big), \tag{3.7}$$

where we denote by $\Sigma_t$ the covariance of $v_{t+1} \sim \mathbb{P}(\cdot \,|\, S_t, q_{t+1})$, and the mean concept $\bar{z}_t$ is

$$\bar{z}_t = V_t \big(\mathfrak{K}(K_t, K_t) + \lambda I\big)^{-1} \phi(K_t). \tag{3.8}$$

Combining equation 3.7 and equation 3.8, we can see that $\bar{z}_t \phi(q_{t+1})$ essentially measures the similarity between the query and keys, which is quite similar to the attention mechanism defined in Section 2. However, here the similarity is normalization according to equation 3.8, not by softmax. This motivates us to define a new structure of attention and explore the relationship between the newly defined attention and the original one. For any $q \in \mathbb{R}^{d_k}$, $K \in \mathbb{R}^{t \times d_k}$, and $V \in \mathbb{R}^{t \times d_v}$, we define a variant of the attention mechanism as follows,

$$\mathtt{attn}_\dagger(q, K, V) = V^\top \big(\mathfrak{K}(K, K) + \lambda I\big)^{-1} \mathfrak{K}(K, q). \tag{3.9}$$

From equation 3.7, equation 3.8, and equation 3.9, it holds that the response $v_{t+1}$ for $(t+1)$-th query is distributed as $v_{t+1} \sim N(\texttt{attn}_\dagger(q_{t+1}, K_t, V_t), \Sigma_t)$. We note that $\texttt{attn}_\dagger$ **bakes the BMA algorithm** for the Gaussian linear model **in its architecture**, by first estimating $\bar{z}_t$ via equation 3.8 and deriving the final estimate from the inner product between $\bar{z}_t$ and $q_{t+1}$. Here $\texttt{attn}_\dagger(\cdot)$ is an instance of the *intention mechanism* studied in Garnelo & Czarnecki [2023] and can be viewed as a generalization of linear attention. Recall that we define the softmax attention [Vaswani et al., 2017] for any $q \in \mathbb{R}^{d_k}$, $K \in \mathbb{R}^{t \times d_k}$, and $V \in \mathbb{R}^{t \times d_v}$ as $\texttt{attn}(q, K, V) = V^\top \texttt{softmax}(Kq)$. In the following proposition, we show that the attention in equation 3.9 coincides with the softmax attention as the sequence length goes to infinity.

**Proposition 3.6.** We assume that Assumption 3.5 holds for the feature mapping $\phi$ of Gaussian RBF kernel $\mathfrak{K}_{\texttt{RBF}}$. In addition, we assume that $\|k_t\| = \|v_t\| = 1$. Then, it holds for a constant $C > 0$ that depends on $d_k$ and any $q \in \mathbb{R}^{d_k}$ with $\|q\| = 1$ that $\lim_{T \to \infty} \texttt{attn}_\dagger(q, K_T, V_T) = C \cdot \lim_{T \to \infty} \texttt{attn}(q, K_T, V_T)$.

The proof is in Appendix E.4. Combined with the conditional probability of $v_{t+1}$ in equation 3.7, this proposition shows that **softmax attention approximately encodes BMA** in long token sequences [Wasserman, 2000], and thus is able to perform ICL when prompted after pretraining. This proposition also implies that the output of the attention module contains the information of the hidden concept $z_*$, which will be verified in experiments.

### 3.4 Experimental Validations For Proposition 3.6

We conduct two experiments to verify Proposition 3.6. The first experiment is same as the hidden concept vectors construction experiments in Section 3.2. Proposition 3.6 and equation 3.8 imply that the heads of attention contain the hidden concept information. Thus, we construct the hidden concept vector as the average of the values of twenty selected attention heads. The effectiveness of the constructed hidden concept is demonstrated in Figure 3. This result strongly corroborates with equation 3.8.

In the second experiment, we directly calculate the ratio between $\texttt{attn}_\dagger$ and $\texttt{attn}$. We consider the case $d_v = 1$ and $d_k = d$ for some $d > 0$. The entries in $K$ of equation 3.9 are i.i.d. samples of Gaussian distribution, and the $i-$th entry of $V$ is calculated as the inner product between a Gaussian vector and the $i-$th column. Figure 4 shows the results for $d = 2$ and $d = 3$. It shows that the ratio between $\texttt{attn}_\dagger$ and $\texttt{attn}$ will converge to a constant. This constant depends on the dimension $d$, which originates from Proposition E.5.

## 4 Theoretical Analysis of Pretraining

### 4.1 Pretraining Algorithm

In this section, we describe the pretraining setting. We largely follow the transformer structures in Brown et al. [2020]. The whole network is a composition of $D$ sub-modules, and each sub-module consists of a MHA and a Feed-Forward (FF) fully connected layer. Here, $D > 0$ is the depth of the network. The whole network takes $X^{(0)} = X \in \mathbb{R}^{L \times d}$ as its input. In the $t$-th layer for $t \in [D]$, it first takes the output $X^{(t-1)}$ of the $(t-1)$-th layer as the input and forwards it through MHA with a residual link and a layer normalization $\Pi_{\texttt{norm}}(\cdot)$ to output $Y^{(t)}$, which projects each row of the input into the unit $\ell_2$-ball. Here we take $d_h = d$ in MHA, and the generalization of our result to general cases is trivial. Then the intermediate output $Y^{(t)}$ is forwarded to the FF module. It maps each row of the input $Y^{(t)} \in \mathbb{R}^{L \times d}$ through the same single-hidden layer neural network with $d_F$ neurons, that is $\texttt{ffn}(Y^{(t)}, A^{(t)}) = \texttt{ReLU}(Y^{(t)} A_1^{(t)}) A_2^{(t)}$, where $A_1^{(t)} \in \mathbb{R}^{d \times d_F}$, and $A_2^{(t)} \in \mathbb{R}^{d_F \times d}$ are the weight matrices. Combined with a residual link and layer normalization, it outputs the output of layer $t$ as $X^{(t)}$, that is

$$Y^{(t)} = \Pi_{\texttt{norm}}\big[\texttt{mha}(X^{(t-1)}, W^{(t)}) + \gamma_1^{(t)} X^{(t-1)}\big],$$
$$X^{(t)} = \Pi_{\texttt{norm}}\big[\texttt{ffn}(Y^{(t)}, A^{(t)}) + \gamma_2^{(t)} Y^{(t)}\big]. \tag{4.1}$$

Here we allocate weights $\gamma_1^{(t)}$ and $\gamma_2^{(t)}$ to residual links only for the convenience of theoretical analysis. In the last layer, the network outputs the probability of the next token via a softmax module, that is

$Y^{(D+1)} = \texttt{softmax}(\mathbb{I}_L^\top X^{(D)} A^{(D+1)} / (L\tau)) \in \mathbb{R}^{d_y}$, where $\mathbb{I}_L \in \mathbb{R}^L$ is the vector with all ones, $A^{(D+1)} \in \mathbb{R}^{d \times d_y}$ is the weight matrix, $\tau \in (0,1]$ is the fixed temperature parameter, and $d_y$ is the output dimension. The parameters of each layer are denoted as $\theta^{(t)} = (\gamma_1^{(t)}, \gamma_2^{(t)}, W^{(t)}, A^{(t)})$ for $t \in [D]$ and $\theta^{(D+1)} = A^{(D+1)}$, and the parameter of the whole network is the concatenation of these parameters, i.e., $\theta = (\theta^{(1)}, \cdots, \theta^{(D+1)})$. We consider the transformers with bounded parameters. The set of parameters is

$$\Theta = \Big\{ \theta \,\big|\, \big\|A^{(D+1),\top}\big\|_{1,2} \le B_A, \max\big\{\big|\gamma_1^{(t)}\big|, \big|\gamma_2^{(t)}\big|\big\} \le 1, \big\|A_1^{(t)}\big\|_{\mathbf{F}} \le B_{A,1}, \big\|A_2^{(t)}\big\|_{\mathbf{F}} \le B_{A,2},$$

$$\big\|W_i^{Q,(t)}\big\|_{\mathbf{F}} \le B_Q, \big\|W_i^{K,(t)}\big\|_{\mathbf{F}} \le B_K, \big\|W_i^{V,(t)}\big\|_{\mathbf{F}} \le B_V, t \in [D], i \in [h] \Big\},$$

where $B_A$, $B_{A,1}$, $B_{A,2}$, $B_Q$, $B_K$, and $B_V$ are the bounds of parameter. Here we only consider the non-trivial case where these bounds are larger than 1, otherwise, the magnitude of the output in $D^{\text{th}}$ layer decreases exponentially with growing depth. The probability induced by the transformer with parameter $\theta$ is denoted as $\mathbb{P}_\theta$.

The pretraining dataset consists of $N_{\text{p}}$ independent trajectories. For the $n$-th trajectory with $n \in [N_{\text{p}}]$, a hidden concept $z^n \sim \mathbb{P}_{\mathcal{Z}}(z) \in \Delta(\mathfrak{Z})$ is first sampled, which is the hidden variable of the token sequence to generate, e.g., the theme, the sentiment, and the style. Then the tokens are sequentially sampled from the model in equation 3.2. We view this model as a Markov chain in the *sequence space* $\mathfrak{X}^*$ induced by $z^n$, i.e., $x_{t+1}^n \sim \mathbb{P}(\cdot \,|\, S_t^n, z^n)$ and $S_{t+1}^n = (S_t^n, x_{t+1}^n)$, where $x_{t+1}^n \in \mathfrak{X}$ can be either $r_t$ or $\widetilde{c}_t$ in equation 3.2, and $S_t^n, S_{t+1}^n \in \mathfrak{X}^*$. This Markov chain is defined with respect to the state $S_t^n$, which obviously satisfies the Markov property since $S_i^n$ for $i \in [t-1]$ are contained in $S_t^n$. The pretraining dataset is $\mathcal{D}_{N_{\text{p}}, T_{\text{p}}} = \{(S_t^n, x_{t+1}^n)\}_{n,t=1}^{N_{\text{p}}, T_{\text{p}}}$ where the concepts $z^n$ is hidden from the context and thus unobserved. Here each token sequence is divided into $T_{\text{p}}$ pieces $\{(S_t^n, x_{t+1}^n)\}_{t=1}^{T_{\text{p}}}$. We highlight that this pretraining dataset collecting process subsumes those for GPT, and Masked AutoEncoders (MAE) [Radford et al., 2021]. For GPT, each trajectory corresponds to a paragraph or an article in the pretraining dataset, and $z^n \sim \mathbb{P}_{\mathcal{Z}}(z)$ is realized by the selection process of these contexts from the Internet. For MAE, we take $T_{\text{p}} = 1$, and $S_1^n$ and $x_2^n$ respectively correspond to the image and the masked token.

To pretrain the transformer, we adopt the cross-entropy as the loss function, which is widely used in the training of BERT and GPT. The pretraining algorithm is

$$\widehat{\theta} = \operatorname*{argmin}_{\theta \in \Theta} -\frac{1}{N_{\text{p}} T_{\text{p}}} \sum_{n=1}^{N_{\text{p}}} \sum_{t=1}^{T_{\text{p}}} \log \mathbb{P}_\theta(x_{t+1}^n \,|\, S_t^n). \tag{4.2}$$

We first analyze the population version of equation 4.2. In the training set, the conditional distribution of $x_{t+1}^n$ conditioned on $S_t^n$ is $\mathbb{P}(x_{t+1}^n \,|\, S_t^n) = \int_{\mathfrak{Z}} \mathbb{P}(x_{t+1}^n \,|\, S_t^n, z) \mathbb{P}_{\mathcal{Z}}(z \,|\, S_t^n) \mathrm{d}z$, where the unobserved hidden concept is weighed via its posterior distribution. Thus, the population risk of equation 4.2 is $\mathbb{E}_t[\mathbb{E}_{S_t}[\mathrm{KL}(\mathbb{P}(\cdot \,|\, S_t) \| \mathbb{P}_\theta(\cdot \,|\, S_t)) + H(\mathbb{P}(\cdot \,|\, S_t))]]$, where $t \sim \texttt{Unif}([T_{\text{p}}])$, $H(p) = -\langle p, \log p \rangle$ is the entropy, and $S_t$ is distributed as the pertaining distribution. Thus, we expect that $\mathbb{P}_\theta$ will converge to $\mathbb{P}$. For MAE, the network training adopts $\ell_2$-loss, and we defer the analysis of this case to Appendix F.4.

## 4.2 Performance Guarantee for Pretraining

We first state the assumptions for the pretraining setting.

**Assumption 4.1.** There exists a constant $R > 0$ such that for any $z \in \mathfrak{Z}$ and $S_t \sim \mathbb{P}(\cdot \,|\, z)$, we have $\|S_t^\top\|_{2,\infty} \le R$ almost surely.

This assumption states that the $\ell_2$-norm of the magnitude of each token in the token sequence is upper bounded by $R > 0$. It is satisfied in real applications, where the token is a finite-dimensional vector with bounded components. For example, the tokenizers used in GPT-NeoX-20B [Black et al., 2022] and Llama2 [Touvron et al., 2023] both satisfy this assumption.

**Assumption 4.2.** There exists a constant $c_0 > 0$ such that for any $z \in \mathfrak{Z}$, $x \in \mathfrak{X}$ and $S \in \mathfrak{X}^*$, $\mathbb{P}(x \,|\, S, z) \ge c_0$.

This assumption states that the conditional probability of $x$ conditioned on $S$ and $z$ is lower bounded. This comes from the ambiguity of language, that is, a sentence can take lots of words as its next word. It holds in a wide range of problems, which is verified by the success of LLMs. Concretely, the transformers with finite width, depth, and weights can only model the distribution that satisfies Assumption 4.2., since the last softmax layer of the transformer renders the probability of each token strictly larger than 0. Since transformers have successfully learned a large range of tasks, it is reasonable to assume that those distributions satisfy this assumption. Similar regularity assumptions are also widely adopted in ICL literature [Xie et al., 2021; Wies et al., 2023]. The parameter $c_0$ depends on both the hidden concept set $\mathfrak{Z}$ and the prompt. To state our result, we respectively use $\mathbb{E}_{S \sim \mathcal{D}}$ and $\mathbb{P}_{\mathcal{D}}$ to denote the expectation and the distribution of the average distribution of $S_t^n$ in $\mathcal{D}_{N_\mathrm{p}, T_\mathrm{p}}$, i.e., $\mathbb{E}_{S \sim \mathcal{D}}[f(S)] = \sum_{t=1}^{T_\mathrm{p}} \mathbb{E}_{S_t}[f(S_t)]/T_\mathrm{p}$ for any function $f : \mathfrak{X}^* \to \mathbb{R}$.

**Theorem 4.3.** Let $\bar{B} = \tau^{-1} R h B_A B_{A,1} B_{A,2} B_Q B_K B_V$ and $\bar{D} = D^2 d(d_F + d_h + d) + d \cdot d_y$. Under Assumptions 4.1 and 4.2, the pretrained model $\mathbb{P}_{\hat{\theta}}$ by the algorithm in equation 4.2 satisfies

$$\mathbb{E}_{S \sim \mathcal{D}}\Big[\mathtt{TV}\big(\mathbb{P}(\cdot \mid S), \mathbb{P}_{\hat{\theta}}(\cdot \mid S)\big)\Big]$$
$$= O\bigg(\underbrace{\inf_{\theta^* \in \Theta} \sqrt{\mathbb{E}_{S \sim \mathcal{D}} \mathtt{KL}\big(\mathbb{P}(\cdot | S) \| \mathbb{P}_{\theta^*}(\cdot | S)\big)} + \frac{\sqrt{b^*} t_{\mathrm{mix}}^{1/4} \log 1/\delta}{(N_\mathrm{p} T_\mathrm{p})^{1/4}}}_{\text{approximation error}} + \underbrace{\frac{\sqrt{t_{\mathrm{mix}}}}{\sqrt{N_\mathrm{p} T_\mathrm{p}}}\Big(\bar{D} \log(1 + N_\mathrm{p} T_\mathrm{p} \bar{B}) + \log \frac{1}{\delta}\Big)}_{\text{generalization error}}\bigg)$$

with probability at least $1 - \delta$, where $b^* = \log(\max\{c_0^{-1}, 1 + d_y \exp(B_A/\tau)\})$, and $t_{\mathrm{mix}}$ is the mixing time of the Markov chains induced by $\mathbb{P}$, formally defined in Appendix F.1.

We define the right-hand side of the equation as $\Delta_{\mathrm{pre}}(N_\mathrm{p}, T_\mathrm{p}, \delta)$. The first and the second terms in the bound are the **approximation error**. It measures the distance between the nominal distribution $\mathbb{P}$ and the distributions induced by transformers with respect to KL divergence. If the nominal model $\mathbb{P}$ can be represented by transformers exactly, i.e., the realizable case, these two terms will vanish. The third term is the **generalization error**, and it does not increase with the growing sequence length $T_\mathrm{p}$. This is proved via the PAC-Bayes framework.

This pretraining analysis is missing in most existing theoretical works about ICL. Xie et al. [2021], Wies et al. [2023], and Jiang [2023] all assume access to an arbitrarily precise pretraining model. Although the generalization bound in Li et al. [2023b] can be adapted to the pretraining analysis, the risk definition therein can not capture the approximation error in our result. Furthermore, their analysis cannot fit the maximum likelihood algorithm in equation 4.2. Concretely, their result can only show that the convergence rate of KL divergence is $O((N_\mathrm{p} T_\mathrm{p})^{-1/2})$ with a realizable function class. Combined with Pinsker's inequality, this gives the convergence rate for total variation as $O((N_\mathrm{p} T_\mathrm{p})^{-1/4})$ even in the realizable case.

The deep neural networks are shown to be universal approximators for many function classes [Cybenko, 1989; Yarotsky, 2017]. Thus, the approximation error in Theorem 4.3 should vanish with the increasing size of the transformer. To achieve this, we slightly change the structure of the transformer by admitting a bias term in feed-forward modules, taking $A_2^{(t)} \in \mathbb{R}^{d_F \times d_F}$, and admitting $d_F$ to vary across layers. This mildly affects the generalization error by replacing $D \cdot d_F$ by the sum of $d_F$ of all the layers in Theorem 4.3. We derive the approximation error bound when the dimension of each word is equal to one, i.e., $\mathfrak{X} \subseteq \mathbb{R}$. Our method can carry over the case $d > 1$.

**Proposition 4.4** (Informal). Under certain smoothness conditions, if the parameter space of the transformer function class $\Theta$ is large enough, then for some constant $C > 0$, we have

$$\inf_{\theta^* \in \Theta} \max_{\|S^\top\|_{2,\infty} \le R} \mathtt{KL}\big(\mathbb{P}(\cdot \mid S) \| \mathbb{P}_{\theta^*}(\cdot \mid S)\big) = O\bigg(c_0^{-1} \exp\Big(-\frac{D^{1/4}}{C \cdot B}\Big)\bigg),$$

where $B$ is the parameter related to the smoothness, which is specified in Appendix F.3.

The formal statement and proof are deferred to Appendix F.3. This proposition states that the **approximation error decays exponentially with the increasing depth**. Combined with this result, Theorem 4.3 provides the full description of the pretraining performance.

# 5 ICL Error under Practical Settings

**ICL Average Rrror with an Imperfectly Pretrained Model** In Section 3, we study the ICL average error with a perfect pretrained model. In what follows, we characterize the ICL average error when the pretrained model has an error. Note that the distribution $\mathcal{D}_{\texttt{ICL}}$ of the prompts of ICL tasks can be different from that of pretraining. We impose the following assumption on their relation.

**Assumption 5.1.** We assume that there exists an absolute constant $\kappa > 0$ such that for any ICL prompt $\texttt{pt} \in \mathfrak{X}^*$ with length less and equal to $T_{\rm p}$, it holds that $\mathbb{P}_{\mathcal{D}_{\texttt{ICL}}}(\texttt{pt}) \leq \kappa \cdot \mathbb{P}_{\mathcal{D}}(\texttt{pt})$.

This assumption states the coverage of the prompt distribution by the pretraining distribution. We note that there will be an information-theoretic barrier without this assumption. For example, if the pretraining data does not contain any material about a specific mathematical symbol in the ICL prompt. In this case, it will be extremely difficult for the LLM to derive the correct prediction, since the meaning of this math symbol is unclear to LLMs. The parameter $\kappa$ scales with the size of the hidden space $\mathfrak{Z}$ and the length of the prompt. In the worst case, $\kappa = O(|\mathfrak{Z}| * T_{\rm p})$. Usually, we note that $N_{\rm p}$ is substantially larger than $\kappa$. Concretely, in the pretraining data generation process, each hidden concept will generate a large number of sentences. The real-world data set contains a large number of sentences explaining the same concept, e.g., the data from Wikipedia. We then have the following theorem characterizing the ICL average error of the pretrained model.

**Theorem 5.2** (ICL average error of Pretrained Model)**.** We assume that the underlying hidden concept $z_*$ maximizes $\sum_{i=1}^{t} \log \mathbb{P}(r_i \,|\, \texttt{pt}_{i-1}, z)$ for any $t \in [T]$ ($T \leq T_{\rm p}$) and there exists an absolute constant $\beta > 0$ such that $\log(1/p_0(z_*)) \leq \beta$. Under Assumptions 3.1, 4.1, 4.2, and 5.1, we have with probability at least $1 - \delta$ that

$$T^{-1}\sum_{t=1}^{T} \mathbb{E}_{\texttt{pt}\sim\mathcal{D}_{\texttt{ICL}}}\Big[\log \mathbb{P}(r_t \,|\, z_*, \texttt{pt}_{t-1}) - \log \mathbb{P}_{\widehat{\theta}}(r_t \,|\, \texttt{pt}_{t-1})\Big] \leq \mathcal{O}\big(\beta/T + \kappa \cdot b^* \cdot \Delta_{\rm pre}(N_{\rm p}, T_{\rm p}, \delta)\big),$$

where we denote by $\Delta_{\rm pre}(N_{\rm p}, T_{\rm p}, \delta)$ the pretraining error in Theorem 4.3, and $b^* = \log \max\{c_0^{-1}, 1 + d_y \exp(B_A/\tau)\}$.

We note that $d_y$ and $B_A$ are the parameters of transformers defined in Section 4.1. The requirement $T \leq T_{\rm p}$ originates from that the pertaining process can only guarantee the performance for prompt not longer than $T_{\rm p}$. Theorem 5.2 shows that the expected ICL average error for the pretrained model is upper bounded by the sum of two terms: **(a) the ICL average error for the underlying true model** and **(b) the pretraining error**. These two terms are separately bounded in Sections 3 and 4.

**Prompting with Wrong Input-Output Mappings** In the real-world implementations of ICL, the provided input-output examples may not conform to the nominal distribution induced by $z_*$, and the outputs in examples can be *perturbed*. The perturbed prompt is then denoted as $\texttt{pt}' = (S'_t, \widetilde{c}_{t+1})$, where $S'_t = (\widetilde{c}_1, r'_1, \cdots, \widetilde{c}_t, r'_t) \in \mathfrak{X}^*$, and $r'_i$ for $i \in [t]$ is the modified output. Then we have that

$$\mathbb{E}_{\texttt{pt}'}\Big[\text{KL}\big(\mathbb{P}(\cdot \,|\, \widetilde{c}_{t+1}, z_*)\|\mathbb{P}_{\widehat{\theta}}(\cdot \,|\, S'_t, \widetilde{c}_{t+1})\big)\Big] = \mathcal{O}\bigg(\kappa\Delta_{\rm pre}(N_{\rm p}, T_{\rm p}, \delta) + \exp\bigg(-\frac{\sqrt{t} \cdot \Delta_{\rm KL}}{2(1+l)\log 1/c_0}\bigg)\bigg),$$

where $\Delta_{\rm KL}$ is a distinguishbility parameter, and $l$ is the maximal length of $\widetilde{c}_t$. The first term is the pretraining error in Theorem 4.3, which is related to the size of the pretraining set and the capacity of the neural networks. The second term is the ICL error. Intuitively, this term represents the concept identification error. If the considered task $z_*$ is distinguishable, i.e., satisfying Assumption G.3, this term decays to 0 exponentially in $\sqrt{t}$. The required assumptions and formal statement are in Appendix G.2.

# 6 Conclusion

In this paper, we investigated the theoretical foundations of ICL for the pretrained language models. We proved that the perfectly pretrained LLMs implicitly implements BMA with regret $\mathcal{O}(1/t)$ over a general response generation modeling, which subsumes the models in previous works. Based on this, we showed

that the attention mechanism parameterizes the BMA algorithm. Analyzing the pretraining process, we demonstrated that the total variation between the pretrained model and the nominal distribution consists of the approximation error and the generalization error. The combination of the ICL regret and the pretraining performance gives the full description of ICL ability of pretrained LLMs. We mainly focus on the prompts that comprise several examples in this work and leave the analysis of instruction-based prompts for future works.

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
