# OpenReview forum: "What and How does In-Context Learning Learn? Bayesian Model Averaging, Parameterization, and Generalization"
_TMLR — Rejected by TMLR_

### Review · Reviewer_efA6 · 2024-06-21

**Summary Of Contributions:**

### Summary

This paper presents a theoretical exploration of In-Context Learning (ICL) in Large Language Models (LLMs) through the lens of Bayesian Model Averaging (BMA). The authors argue that LLMs, particularly those utilizing transformer architectures, are inherently capable of performing BMA during the ICL process. This capability enables them to effectively generalize from limited data and few-shot examples without the need for explicit retraining or parameter adjustments.

**Audience:**

Yes

**Broader Impact Concerns:**

### Broader Impact Concerns

I have no concerns regarding the ethical implications of the work. The paper focuses on theoretical aspects of In-Context Learning in Large Language Models, which are primarily mathematical and technical in nature.

**Claims And Evidence:**

Yes

**Requested Changes:**

### Requested Changes

To further strengthen the manuscript, I would suggest the following revisions:

1. **Empirical Validations:** Provide more discussions or experiments in practical settings to better validate the theoretical claims, such as the ICL averaged error.
2. **Assumptions:** Offer a more in-depth analysis of each assumption, especially regarding their validity in practical settings.
3. **Formatting Adjustments:** Increase the spacing between the captions of Figures 2-4 or adjust their format to enhance readability.
4. **Expanding Related Work:** Include a broader discussion of related work.

**Strengths And Weaknesses:**

### Strengths

1. **Theoretical Insights:** The paper articulates a series of well-formulated propositions and theorems that describe how BMA can be implemented within LLMs. It includes bounds on the ICL averaged error rate, which deepens our understanding of ICL in LLMs.
2. **Empirical Verification:** The authors conduct empirical experiments in controlled toy settings, such as pretraining transformers for regression tasks, which provide experimental evidence to support some of the theoretical claims.
3. **Timeliness:** The paper addresses one of the most pressing challenges in AI research—the ability of models to learn effectively from a few examples. This is highly relevant given the widespread adoption of these models in practical applications.

### Weaknesses

1. **Empirical Validations:** The empirical evaluations are mainly in toy settings, which are somewhat removed from real-world scenarios. It would be beneficial to verify some of the theoretical claims, such as the ICL averaged error, in more practical settings.
2. **Assumptions:** While the paper discusses the implications and meanings of its foundational assumptions, a more in-depth analysis of whether these assumptions hold in real-world LLMs would strengthen the paper.
3. **Formatting:** Figures 2-4 are slightly squeezed together, especially their captions, which could impact readability.
4. **Related Work:** The discussion of related work could be expanded. For instance, the study by Mikulik et al. on Meta-trained agents implementing Bayes-optimal agents at NeurIPS 2020 could provide relevant insights.

---

> ### Author Response · Authors · 2024-07-15
> **Rebuttal to Reviewer efA6 Part 1/2**
>
> We thank the reviewer for the valuable feedback. We address the major concerns in the following.
> ___
> **Concern 1 (Practical Relevance of Experiments)**
>
> We would like to highlight that our experimental settings have two types of prompts: synthetic prompts and natural language prompts. These two kinds of prompts are adopted to verify our theoretical results **quantitatively** and **qualitatively**. We note that both types of prompts are practical and widely adopted in the existing works.
>
> * **(Natural Language ICL Experiments)** This experiment explicitly constructs the hidden concept derived in LLMs by averaging the values of the selected attention heads. We note that the **natural language** examples in the experimental prompts are **highly similar** to the examples in the real scenarios. Similar settings are adopted in [1,2]. They also adopt the input-output pairs in the natural language form as the prompts to test the ICL ability of LLMs. The experimental results verify that the LLM can explicitly construct the hidden concept at attention heads, which is well supported by Propositions 3.3 and 3.6.
>
> * **(Least Square ICL Experiments)** This experiment tests the ICL ability of LLMs on least square examples. As a representative synthetic experimental setting, this experiment enables us to **quantitatively** measure the ICL performance, which is widely studied in [3,4]. These existing works focus on the least square problem to study the ICL behavior of LLMs. Thus, we also conduct experiments in this setting, and our model is based on GPT-2. By calculating the size of the ball in the $d-$dimensional space, we note that the experimental results are strongly corroborated with Proposition 3.4.
>
> * **(Function Value Prediction Experiments)** This experiment demonstrates the ICL ability of LLM on the function value prediction task. This task takes the value of different functions as examples in the prompt and lets LLM predict the next value, which is adopted in [5] as a **representative setting** for time-series related problems. Our experimental results show that the cumulative negative log-likelihood (multiplied ICL average error) remains bounded in a large range, which verifies Proposition 3.4
>
> * **(Attention Limit Experiments)** This experiment directly calculates the ratio between the newly defined attention in (3.9) and the original attention. The queries, keys, and values are i.i.d. samples of **Gaussian** random variables, whose support is the whole space. The experimental results verify Proposition 3.6.
>
> * **(Bandit Experiments)** This experiment demonstrates the decision-making ability of LLM from historical data. Similar **decision-making agents** are studied in [6,7]. We note that LLMs can exploit the special structure in the examples to make the correct decision, which is attributed to learning the hidden concept in the distribution space. This corroborates with Proposition 3.3.
> ___
>
>
> **Concern 2 (Assumption Analysis)**
> We would like to discuss the practice-relevance of our assumptions as follows:
>
> * **Hidden concept model (Assumptions 3.1, 3.5, and G.1)** These assumptions mainly assume that the examples in ICL can be generated from a **hidden concept model**. Concretely, this assumption assumes that the examples convey a **main semantic** $z$, e.g., ''finer taxonomy'' in Figure. 6. Then the response is generated from the hidden variable $h$, e.g., ''class'', and covariate, e.g., ''otter'' in Figure 6. This model is general and includes the models in the previous ICL works [8,9,10], where they all assume that hidden variable $h$ cannot change across examples. Assumption G.1 is used in a more difficult setting where prompts are perturbed, so it requires a stronger condition $h_t=z$ and degenerates to the settings in [8,9,10].
>
>     For Assumption 3.5, which is a special case of Assumption 3.1, we would like to highlight that Eqn. (3.6) is general. The generality comes from two sources: (i) two learnable mappings $v_{\ast}(\cdot)$ and $k_{\ast}(\cdot)$ are **neural networks** and are general enough to represent continuous functions. (ii) the feature mapping $\phi$ and the corresponding RKHS make Kernel Mean Embedding (KME) have **sufficient expressiveness**, since all the operations on distribution can be captured by some operations in KME space.
>
> * **Perfect pretraining(Assumption 3.2)** This assumption assumes that LLMs are perfectly trained. We present this assumption in Section 3 only for the conciseness. This assumption is **replaced** later by analyzing the **pretraining error** in Section 4.

---

> ### Author Response · Authors · 2024-07-15
> **Rebuttal to Reviewer efA6 Part 2/2**
>
> * **Properties of language(Assumptions 4.1 and 4.2)**: These assumptions state the properties of language and its embedding. Assumption 4.1 assumes that all the tokens are embedded to vectors with finite $l_{2}$ norm. This is naturally satisfied by **all the LLMs**[11,12], since the tokenizers of LLMs always produce vectors with **finite** dimensions and values. Assumption 4.2 states the lower boundness of the nominal distribution. data. This comes from the **ambiguity** of language, that is, a sentence can take lots of words as its next word. In addition, the transformers with finite width, depth, and weights can **only model** the distribution that satisfies Assumption 4.2., since the last softmax layer of transformers renders the probability of each token **strictly larger than $0$**. Since transformers have successfully learned a large range of tasks, it is reasonable to assume that those distributions satisfy this assumption. Similar regularity assumptions are also widely adopted in ICL literature [8,13].
>
> * **Data coverage(Assumptions 5.1 and G.2)**: These assumptions mainly states that the distribution of the ICL prompt and the hidden concept is covered by that of the pretraining data. We note that there will be an information-theoretic barrier without this assumption, the ICL learning problem presents an **information-theoretic barrier**. For example, if the pretraining data does not contain any material about a specific mathematical symbol in the ICL prompt. In this case, it will be extremely difficult for the LLM to derive the correct prediction, since the meaning of this math symbol is unclear to LLMs.
>
> * **Sufficient information in perturbed distribution (Assumptions G.3 and G.4)**: These assumptions state that the perturbed prompt is distinguishable for LLMs to identify the correct task. If the **distinguishability** does not hold, there will be another concept closer to the perturbed distribution than the true concept. Then LLMs will predict according to the wrong concept with higher probability. The independence of examples guarantees that the information **cumulates** when more examples are provided.
>
> ___
>
> **Concern 3 (Formatting and Related Works)**
>
> For the formatting, we will adjust the captions of Figures 2-4 in the revised version.
>
> For the related work, we will add the reference [14] for the insights related to the Bayes-optimal learner.
>
> ___
> **Reference**
>
> [1] Hendel, Roee, Mor Geva, and Amir Globerson. "In-context learning creates task vectors." arXiv preprint arXiv:2310.15916 (2023).
>
> [2] Todd, Eric, et al. "Function vectors in large language models." arXiv preprint arXiv:2310.15213 (2023).
>
> [3] Garg, Shivam, et al. "What can transformers learn in-context? a case study of simple function classes." Advances in Neural Information Processing Systems 35 (2022): 30583-30598.
>
> [4] Akyürek, Ekin, et al. "What learning algorithm is in-context learning? investigations with linear models." arXiv preprint arXiv:2211.15661 (2022).
>
> [5] Gruver N, Finzi M, Qiu S, et al. Large language models are zero-shot time series forecasters[J]. Advances in Neural Information Processing Systems, 2024, 36.
>
> [6] Wang, Guanzhi, et al. "Voyager: An open-ended embodied agent with large language models." arXiv preprint arXiv:2305.16291 (2023).
>
> [7] Wang, Lei, et al. "A survey on large language model based autonomous agents." Frontiers of Computer Science 18.6 (2024): 186345.
>
> [8] Xie S M, Raghunathan A, Liang P, et al. An explanation of in-context learning as implicit bayesian inference[J]. arXiv preprint arXiv:2111.02080, 2021.
>
> [9] Wang X, Zhu W, Saxon M, et al. Large language models are implicitly topic models: Explaining and finding good demonstrations for in-context learning[C]//Workshop on Efficient Systems for Foundation Models@ ICML2023. 2023.
>
> [10] Jiang H. A latent space theory for emergent abilities in large language models[J]. arXiv preprint arXiv:2304.09960, 2023.
>
> [11] Achiam J, Adler S, Agarwal S, et al. Gpt-4 technical report[J]. arXiv preprint arXiv:2303.08774, 2023.
>
> [12] Touvron H, Martin L, Stone K, et al. Llama 2: Open foundation and fine-tuned chat models[J]. arXiv preprint arXiv:2307.09288, 2023.
>
> [13] N. Wies, Y. Levine, A. Shashua. The learnability of in-context learning. arXiv preprint arXiv:2303.07895 (2023).
>
> [14] Mikulik, Vladimir, et al. "Meta-trained agents implement bayes-optimal agents." Advances in neural information processing systems 33 (2020): 18691-18703.

---

> > ### Comment · Reviewer_efA6 · 2024-08-19
> >
> > ### Empirical Validations
> >
> > I appreciate the authors' comprehensive response addressing the practical relevance of the experiments. The detailed explanations of both synthetic and natural language prompts demonstrate a robust approach to validating the theoretical results across diverse settings. It's good to know that many of these experimental settings are widely adopted in the field. By inspecting the example prompts provided in the figures in the main text for ICL experiments and the table in the appendix for the bandit experiments, I agree that the prompts closely resemble practical applications.
> >
> > ### Assumption Analysis
> >
> > I appreciate the clarifications provided on the assumptions underlying your model. For Assumptions 3.1 and 3.5 regarding the hidden concept model, I agree that in the ICL setting, the in-context examples typically convey a single hidden concept, such as following instructions or answering questions in a specific format. Treating neural network mappings as RKHS feature maps also seems a reasonable assumption given the nature of the tasks.
> >
> > Assumptions 3.2 (perfect pretraining) and 4.1 (properties of language and embedding) are plausible and align well with the operational characteristics of LLMs. Regarding Assumption 4.2, although I'm cautious about the generality of the assumption that the conditional probability for any token is always lower bounded, I recognize that it is a widely used approximation in the literature.
> >
> > For Assumption 5.1 on data coverage, it seems reasonable to expect that the distribution of ICL prompts and the hidden concept is covered by the pretraining data. However, I wonder if it's possible to alleviate this assumption somewhat, given LLMs’ ability to generalize to unseen hidden concepts in the training data.

---

> > > ### Author Response · Authors · 2024-08-20
> > >
> > > We thank the reviewer for the valuable feedback. We address the major concerns in the following.
> > > ___
> > >
> > > **(Justification of Assumption 4.2)**
> > >
> > > We would like to justify Assumption 4.2 via: (i) the success of LLMs on a large range of tasks (ii) the mild dependency on the $c_0$ and assumptions in the existing works.
> > >
> > > First, the transformers with finite width, depth, and weights can **only model** the distribution that satisfies Assumption 4.2., since the last softmax layer of transformers renders the probability of each token **strictly larger than $0$**. Since transformers have successfully learned a large range of tasks, it is reasonable to assume that those distributions satisfy this assumption.
> > >
> > > Second, we note that the bound in Theorem 4.3 depends on $c_{0}$ in Assumption 4.2 as $O(\log(c_{0}^{-1}))$. Thus, the bound in Theorem 4.3 depends on $c_{0}$ **mildly**. In addition, similar regularity assumptions are **widely accepted** in the existing works, including Assumption 5 in [1] and Assumption 3 in [2].
> > >
> > > ___
> > >
> > >
> > > **(Concern About Data Coverage)**
> > >
> > > We would like to explain the data coverage assumption from two perspectives.
> > >
> > > First, we note that the data coverage is required for the efficient generalization. As mentioned in the previous response, there exists an **information-theoretic barrier** when the distribution of the prompt in the inference stage is **not covered** by that in the training stage. For example, the LLMs never trained on the data with math equations will have poor performance in math in the inference stage.
> > >
> > > Second, we note that there exist some ways to relax this assumption. In fact, our manuscript focuses on the multi-layer transformer and does not analyze the value of learned weights. However, some existing works derive the **closed-form solution** of the trained weights of shallow transformers under a specfic training method [3]. When the weights demonstrate some **causal structure**, we can relax this data coverage assumption [3,4]. In addition, if the learned weights contain certain **invariant property**, it is also possible to relax the data coverage assumption [4].
> > >
> > > ___
> > >
> > > **Reference**
> > >
> > > [1] Xie S M, Raghunathan A, Liang P, et al. An explanation of in-context learning as implicit bayesian inference[J]. arXiv preprint arXiv:2111.02080, 2021.
> > >
> > > [2] Wies N, Levine Y, Shashua A. The learnability of in-context learning[J]. Advances in Neural Information Processing Systems, 2024, 36.
> > >
> > > [3] Nichani E, Damian A, Lee J D. How transformers learn causal structure with gradient descent[J]. arXiv preprint arXiv:2402.14735, 2024.
> > >
> > > [4] Liu J, Shen Z, He Y, et al. Towards out-of-distribution generalization: A survey[J]. arXiv preprint arXiv:2108.13624, 2021.

---

### Review · Reviewer_UiHW · 2024-07-06

**Summary Of Contributions:**

This mainly theoretical work mathematically analyses in-context learning in transformer-based large language models. The work proves that, under somewhat strict assumptions that the in-context learning (ICL) distribution follows a latent variable model and the model is perfectly pretrained, that ICL performs Bayesian Model Averaging. It further shows error bounds of the average ICL error. Furthermore, the work analyses how attention is an inductive bias that enables ICL. Lastly, the work relaxes the assumption of a perfectly pretrained model, and analyses the average ICL error in this setting, again providing an upper bound.

This work presents theoretical results which are highly related to previous work (for instance Xie et al.), but the results are to the best of my knowledge novel. The theoretical results are augmented by few, yet very interesting experimental results which nicely illustrate them and give a glimpse at their potential practical relevance.

**Audience:**

Yes

**Broader Impact Concerns:**

I have no concerns on ethical implications of the work.

**Claims And Evidence:**

Yes

**Requested Changes:**

- Regarding the concerns expressed in Section 3, it would be important to highlight or include a discussion to what degree Assumptions 3.1 and 3.2 are realistic. -- Furthermore, any statements on ICL implementing BMA should be made more precise. Sentences like at the top of Section 3 “First, we show that the statistical model of ICL and the training loss enable LLMs to implement BMA for ICL.” should add that this only holds under the stated assumptions; as is such statements are overclaiming their impact. -- Addressing this point and the discussion is important for my final recommendation.

- The result on validating Prop. 3.4 empirically is very interesting, but the presented empirical evidence is somewhat limited. -- Could this experiment be repeated under a different task, and could it likewise be shown that significant error only presents up to $T \leq d$?

- I don’t precisely understand how the hidden concept vectors in the exp. validation of Prop 3.6 are constructed. Could you please elaborate?

- In my view, the interesting experiments which augment the theory could be further extended. They are very limited, but very interesting. This would make the work significantly stronger, but I don’t expect such changes during this rebuttal.

- Could you please explain Figure 4?


Small issues:

- p. 5: “Hiddn Markov Model (HMM)”

- Assumption 3.2: “A LLM” -> “An LLM”

- Figure 2: I suggest having a single caption for all subfigures or insert more space

- p. 11: “ICL Average Rrror”

**Strengths And Weaknesses:**

Strengths:

- The theoretical results are clearly motivated and interesting. Assumptions (even though very strict (see weaknesses)) are clearly stated.

- I find the experiments to empirically validate each theoretical proposition insightful and interesting to the community. I particularly enjoyed the insight of conditioning on hidden vectors constructed from attention heads in 3.4 and Figure 3.

- The work nicely compares against related work. An example is that the authors are aware and carefully contrast to Xie et al., stating that in their assumptions, the hidden variables for each ICL example can depend on each other, while in Xie et al., another influential work analysing similar research questions, the hidden variables are assumed to be independent.

- The paper is well-written, the figures are (with some exceptions pointed out below) clear and interesting.


Weaknesses:

- Assumptions 3.1 and 3.2, which are key assumptions for the two main Propositions in Section 3 (Prop. 3.3 and 3.4), both in itself are somewhat unrealistic. Neither the response and covariate variables follow a hidden variable model in practice (if so, what is it?), nor is an LLM perfectly pretrained (as we have finitely many samples, for instance). -- The authors criticised previous work for ““mak[ing] unrealistic assumptions on the pretrained models”, however, particularly 3.2 is equally unrealistic. This weakens for me the relevance of the theoretical results in Section 3 and their interest for the TMLR community. -- It is worth noting that Assumption 3.2 is relaxed in Section 4 onwards, and particularly Section 5 makes a strong attempt to make the theoretical results practically relevant which I appreciate. It is also worth noting that these concerns are to some degree alleviated by the insightful experiments which support and augment the theory. -- Similar questions arise for Assumption 3.5.

- The theoretical results are sometimes hard to understand. Further explanation and intuition of the results would be useful, particularly in the surrounding paragraphs.

- On Prop 3.4: I struggle to see its relevance in practice. Here, we would not know the prior probability P_Z(z_\*) of the latent concepts. -- Can you explain which insight 3.4 yields?

---

> ### Author Response · Authors · 2024-07-15
> **Rebuttal to Reviewer UiHW Part 1/3**
>
> We thank the reviewer for the valuable feedback. We address the major concerns in the following.
> ___
> **Concern 1(Practical Relevance of Assumptions 3.1 and 3.2)**
>
> We would like to discuss the practical relevance of Assumptions 3.1 and 3.2 in the following.
>
> * **(Assumption 3.1)** We would like to discuss Assumption 3.1 from two perspectives.
>
>     First, the **hidden concept model** in Assumption 3.1 holds in a wide range of prompts. Concretely, this assumption assumes that the examples in a prompt convey a **main semantic** $z$, e.g., ''finer taxonomy'' in Figure. 6. Then the response is generated from the hidden variable $h$, e.g., ''class'', and covariate, e.g., ''otter'' in Figure 6. In the real-world applications, the prompts always contain clear semantics. In addition, this assumption does **not** put any constraint on the form of $f$ and $g _{z^{\ast}}$, which renders it **general** enough to model the complex functions in realistic scenarios. Similar models are also widely accepted in the linguistic community [1,2].
>
>     Second, the data model in this assumption is general and **includes** the models in the previous works [3,4,5], where they all assume that hidden variable $h$ **cannot** change across examples. In contrast, our model admits the hidden variable $h$ to evolve according to (3.1).
>
> * **(Assumption 3.2)** We present Assumption 3.2 in Section 3 only for the conciseness of the presentation. In fact, the ICL learning error can be decomposed into two parts: **prompting error of perfectly pretrained model**, and the **pretraining error**, i,e.,
> \begin{align}
>     \text{err} _{\mathrm{ICL}}\leq \text{err} _{\mathrm{prompting}} + \text{err} _{\mathrm{pretraining}}.
> \end{align}
> In Section 3, we focus on the first part and thus require the perfect pretraining assumption. In Section 4, we analyze the pretraining error, which consists of the generalization error and the approximation error. In Section 5, we **remove Assumption 3.2** by combining the results in Sections 3 and 4.
> ____
> **Concern 2(Explanations and Intuitions of Results)**
>
> We would like to provide the explanations and intuitions of each theoretical result.
>
> * **(Proposition 3.3)** This proposition states that the prefectly pretrained model will perform **BMA algorithm** under the data model in Assumption 3.1. With BMA algorithm, the LLMs implicitly construct an **estimate of the hidden concept $z$** and use the posterior of this concept to make the prediction. Intuitively, this implies that the ICL error can be reflected in the hidden concept space (Proposition 3.4), and it is interesting to investigate why LLMs are able to construct the hidden concept (Proposition 3.6).
>
> * **(Proposition 3.4)** This assumption derives the bounds of the ICL average error. It states the average ICL error of the perfectly pretrained LLM is upper bounded by $t^{-1}\log (\mathbb{P} _{\mathcal{Z}}(z _{\ast}))^{-1}$. As discussed in Proposition 3.3, this result represents the ICL error in the hidden concept space. The bound is also intuitive. Concretely, if the prior $\mathbb{P} _{\mathcal{Z}}(z _{\ast})$ for the concept $z _{\ast}$ is small, i.e., this concept appears in the pretraining corpus with **small probability**, then the LLMs will incur **large ICL error** on this concept.
>
> * **(Proposition 3.6)** This proposition shows that the attention mechanism is able to implement **BMA algorithm** under the Gaussian linear model (Assumption 3.5). Intuitively, the output of the attention model is a **weighted sum** of values with weights proportional to the exponential inner product between keys and query. This regression behavior makes it able to estimate the nominal hidden concept $z$ and use it for the prediction.
>
> * **(Theorem 4.3 and Proposition 4.4)** These results jointly provide the pretraining error analysis of the LLMs. These results state that the pretraining error consists of two parts: the **generalization error** and the **approximation error**. The generalization error scales mildly with the length of the context $T _{\mathrm{p}}$ and scales as $O(N _{\mathrm{p}}^{-1/2})$ with respect to the number of samples. This supports the fact that LLMs are able to process long context. The approximation error exponentially decreases with the increasing depth. This originates from the transformer structure.
>
> * **(Theorem 5.2)** This result combines the ICL error of the perfectly pretrained model and the pretraining error. Intuitively, the ICL error of the pretrained model comes from two perspectives: (i) the examples in the prompt may not be informative enough to make the LLMs predict precisely (ii) the training of the LLMs does not find the optimal solution, i.e.,
> \begin{align}
>     \text{err} _{\mathrm{ICL}}\leq \text{err} _{\mathrm{prompting}} + \text{err} _{\mathrm{pretraining}}.
> \end{align}
> We note that even the **perfectly pretrained model** suffers from the first error source. Theorem 5.2 derives the error as the sum of these two kinds of errors.

---

> > ### Comment · Reviewer_UiHW · 2024-07-17
> > **Thank you!**
> >
> > I appreciate the detailed discussion on the review!
> >
> > On Assumption 3.1, I would like to add that there is recent evidence against a latent concept in this paper [1], while many other works also in the LLM and Bayesian context make a similar assumption. Discussing this could be a useful addition to the paper. The latent concept assumption can sometimes also be practically useful [2].
> >
> > [1] https://arxiv.org/pdf/2406.00793
> > [2] https://www.nature.com/articles/s41586-024-07421-0

---

> ### Author Response · Authors · 2024-07-15
> **Rebuttal to Reviewer UiHW Part 2/3**
>
> **Concern 3 (Insights of Proposition 3.4)**
>
> We would like to explain the insights of Proposition 3.4 from two perspectives.
>
> First, we can view the result from the perspective of **online learning**. We consider the quantity $\mathrm{Reg}(t) = \sum _{i=1}^{t} \log \mathbb{P}(r _{i} |  \mathrm{pt} _{i-1}, z _{\ast}) - \sum _{i= 1}^t \log \mathbb{P} _{\theta}(r _{i} |  \mathrm{pt} _{i-1})$, which is known as the regret in the online learning community. We note that Proposition 3.4 implies that $\mathrm{Reg}(t)\leq \log (\mathbb{P} _{\mathcal{Z}}(z _{\ast}))^{-1}$. This regret is **independent** of $t$. This **constant regret** bound implies that the ICL error rate of LLMs is fast, which is attributed to the BMA algorithm implemented by the LLMs.
>
> Second, we note that the bound $t^{-1}\log (\mathbb{P} _{\mathcal{Z}}(z _{\ast}))^{-1}$ increases if the prior probability $\mathbb{P} _{\mathcal{Z}}(z _{\ast})$ decreases. Intuitively, the LLMs are difficult to predict precisely on the concept $z _{\ast}$ that appears with **small probability** in the pretraining data. For example, here the hidden concept $z _{\ast}$ can be solving complex math equations or writing long codes, which is rarely covered by the training of GPT-2. Thus, it will be hard for the GPT-2 to perform these two tasks without fine-tuning.
>
> ___
>
> **Concern 4 (Statements of the Claims)**
>
> Thanks to the reviewer for the suggestions. We will add the assumptions in the statements of our claims.
>
> ___
>
> **Concern 5 (Empirical Validation of Proposition 3.4)**
>
> We note that our result in Proposition 3.4 implies that the $Reg(t) = \sum _{i=1}^{t} \log \mathbb{P}(r _{i} |  \mathrm{pt} _{i-1}, z _{\ast}) - \sum _{i=1}^t \log \mathbb{P} _{\theta}(r _{i} |  \mathrm{pt} _{i-1})$ can be **upperbounded by a constant**. In the time series experiment, which is included in Appendix C.2, we consider the prediction of a deterministic function. We have that $\mathbb{P}(r _{i} |  \mathrm{pt} _{i-1}, z _{\ast}) =1$, since the function is deterministic. Figure 8 shows the values of $\text{CNLL} _{t}=-\sum _{i=1}^{t}\log\hat{\mathbb{P}}(r _{i}\,|\,\mathrm{pt} _{i-1})$. The results show that the cumulative negative log-likelihoods are stepped, which means that the cumulative negative log-likelihoods are **upper bounded by constants in a long period**. This corroborates with Proposition 3.4.
>
> As mentioned by the reviewer, the significant error only presents up to $T\leq d$ in the least square prompts. However, such phenomena may **not be universal** in general tasks. For example, for the problem involving natural language, it is hard to properly **define the dimension $d$** for this problem. In our work, we focus on the theoretical understanding of ICL under the **general** assumptions. Thus, we leave the finer-grained analysis for the least square ICL problem as the future work.
>
> ___
>
> **Concern 6 (Explanation of the Hidden Concept Vector Construction)**
>
> We would like to explain the details of the hidden concept vector construction. We follow the main ideas in [6,7]. Given the prompts generated from the same hidden concept $z$, we calculate the **average value** of each attention head by prompting the LLMs with different prompts and averaging the attention head values. Then we select the attention head according to its **average indirect effect**. The hidden concept vector is the sum of the average value of the selected attention heads.
>
> ___
>
> **Concern 7 (Explanations of Figure 4)**
>
> In Figure 4, we empirically verify Proposition 3.6. This proposition states that with Assumption 3.5, the outputs of the softmax attention and the attention defined in (3.9) are **same up to a constant** when the number of examples increases to infinity. The attention in (3.9) corresponds to the output of the BMA algorithm. In Figure 4, we sample $c_t$ from the Gaussian distribution and directly compute the **ratio** between these two attention mechanisms. The experimental results show that the ratio between them gradually converges to a constant as the number of examples increases, which corroborates with Proposition 3.6.
>
> ___

---

> > ### Author Response · Authors · 2024-07-15
> > **Rebuttal to Reviewer UiHW Part 3/3**
> >
> > **Reference**
> >
> > [1] Andrews M, Vigliocco G. The hidden Markov topic model: A probabilistic model of semantic representation[J]. Topics in Cognitive Science, 2010, 2(1): 101-113.
> >
> > [2] Rahman M M, Wang H. Hidden topic sentiment model[C]//Proceedings of the 25th international conference on world wide web. 2016: 155-165.
> >
> > [3] Xie S M, Raghunathan A, Liang P, et al. An explanation of in-context learning as implicit bayesian inference[J]. arXiv preprint arXiv:2111.02080, 2021.
> >
> > [4] Wang X, Zhu W, Saxon M, et al. Large language models are implicitly topic models: Explaining and finding good demonstrations for in-context learning[C]//Workshop on Efficient Systems for Foundation Models@ ICML2023. 2023.
> >
> > [5] Jiang H. A latent space theory for emergent abilities in large language models[J]. arXiv preprint arXiv:2304.09960, 2023.
> >
> > [6] Hendel, Roee, Mor Geva, and Amir Globerson. "In-context learning creates task vectors." arXiv preprint arXiv:2310.15916 (2023).
> >
> > [7] Todd, Eric, et al. "Function vectors in large language models." arXiv preprint arXiv:2310.15213 (2023).

---

> ### Author Response · Authors · 2024-07-26
>
> Thank the reviewer for the reply! We note that the exploration of the latent concept assumption in the LLM community is a hot topic, as mentioned by the reviewer. The authors of [1] demonstrate that the pretrained LLMs do not perform the **exact Bayesian inference** with the **exchangeable** data by showing the violation of the martingale property on some tasks, including the prediction of random variables and diseases. We would like to highlight that this **does not contradict** with our results. First, our data model in Assumption 3.1 is a general **dynamic** data model among the ICL examples. Our results show that perfectly pretrained LLMs with cross entropy loss implement BMA under this data model. However, [1] does not include the pretraining data model analysis. Second, we show that **pretrained** LLMs implement the **approximate** BMA algorithm for ICL. As shown in Theorem 5.2, when the latent concept assumption holds, the ICL learning error consists of the error from the exact Bayesian inference and the **pretraining error**. The difference between the practical ICL behavior of LLMs and exact Bayesian inference could arise from the pretraining error. For example, the examples in the prompts of [1] is badly covered by the pretraining distribution of LLMs. We will include the discussion of this work in the revised manuscript.
>
> ___
> **Reference**
>
> [1] Falck F, Wang Z, Holmes C. Is In-Context Learning in Large Language Models Bayesian? A Martingale Perspective[J]. arXiv preprint arXiv:2406.00793, 2024.

---

### Review · Reviewer_He72 · 2024-08-07

**Summary Of Contributions:**

This paper aims to find the source of in-context learning of LLMs from a theoretical perspective. Firstly, they show that under certain assumptions, a perfect LLM is essentially performing Bayesian model averaging, which explains its ability to perform ICL. Then they explored the relationship between the attention machanism and ICL ability. They also have a theorem saying that the gap between the learned LLM and the real latent model is bounded under some assumptions to show the rationality of the 'perfect model assumption'.

**Audience:**

Yes

**Broader Impact Concerns:**

No need for a Broader Impact Statement.

**Claims And Evidence:**

Yes

**Requested Changes:**

See Strengths And Weaknesses

**Strengths And Weaknesses:**

The discussed problems are very interesting and important. However, I'm quite confused by some points in this paper:

1. The authors claimed that `existing works fail to explain why LLMs have the ability for ICL'. The authors also claimed that both Xie et al. [2021] and Wies et al. [2023] made unrealistic assumptions, but the only difference we see in the paper is that Xie assume the independence of hidden variables, while Proposition 3.3 in this paper allows hidden variables to depend on each other. (By the way, I don't understand why this is important). The authors also said that both [Xie et al., 2021] and [Wies et al., 2023] have similar assumptions as assumption 4.2 in this paper. The authors should maybe have a more comprehensive comparison between this work and [Xie et al., 2021] / [Wies et al., 2023] to show why their assumptions are more unrealistic. There are other empirical works analyzing the source of ICL ability, such as [Min et al., 2022], which should also be carefully discussed.

2. Isn't proposition 3.3 trivial under the `perfect pretraining' assumption? From my perspective, it's a simple application of the Bayes rule.

3. How can Proposition 3.4 deal with the case where z is a continuous variable? p(z)=0 for all z, which makes the inequality meaningless.

4. I don't think the experiments in this paper are persuasive. For example, in Figure 3, the observed phenomenon is `the LLMs conditioned on the learned hidden concept vectors Pθ(rt+1|ect+1, z∗) have comparable performance with the LLMs prompted with several examples Pθ(rt+1|ptt)'. If proposition 3.3 holds, we will see this phenomenon, but the phenomenon doesn't mean that proposition 3.3 holds.

5. How can we use the conclusions from this work to improve LLMs' ICL ability? I know it's important to understand how things work, but it would be better if we could use the understanding to help improve the current models.

---

> ### Author Response · Authors · 2024-08-12
> **Rebuttal to Reviewer He72 Part 1/2**
>
> We thank the reviewer for the valuable feedback. We address the major concerns in the following.
>
> ___
>
> **(Comparison with Existing Works)**
> We would like to provide a detailed comparison with the existing works that study ICL from the Bayesian view.
>
> |                        	| General Model 	| Evolvable hidden variable in examples 	| Regularity Assumption         	| Convergence rate 	| Pretraining error bound     	|
> |---|---|---|---|---|---|
> | Xie et al. (2021) [1]  	| No (HMM)      	| No                                    	| Yes (Assumptions 1, 2, 5)            	| Yes              	| No                          	|
> | Wies et al. (2024) [2] 	| Yes           	| No (Assumption 2: Near Independence ) 	| Yes (Assumption 3)            	| Yes              	| No                          	|
> | Wang et al. (2023) [3] 	| Yes           	| No                                    	| No                            	| No               	| No                          	|
> | Jiang Hui (2023) [4]   	| Yes           	| No                                    	| Yes ($\varepsilon-$ambiguity) 	| Yes              	| Yes (Only asymptotic result) 	|
> | Ours                   	| Yes           	| Yes                                   	| Yes (Assumptions 4.1, 4.2 )   	| Yes              	| Yes  (non-asymptotic result)                       	|
>
> This table provides a detailed comparison between our work and the existing works. We note that our analysis generalizes the existing works mainly from several perspectives.
>
> **(Model generality)** [1] assumes that the sentences are generated from an **HMM**, where the transition probability of the hidden variable and the emission functions are specified with some assumptions. In contrast, we do **not** assume that there exists a hidden variable such that a token is independent of all the other token conditioning on this hidden variable.
>
> **(Evolvable hidden variable)** We do **not** require that the examples in the prompts are i.i.d. or near i.i.d. In contrast, the examples can depend on each other via evolving hidden variables. This is practice-relevant since people usually come up with examples that share some **inner similarity**.  We highlight that this correlation between examples plays an important role in the language, and this is well-studied in the linguistic community [5,6,7].
>
> **(Perfect pretraining assumption)** We do **not** assume that the LLMs are **perfectly** pretrained by analyzing its pretraining error bound in Theorem 4.3. In contrast, the existing works adopt the perfect pretraining assumption or only provide the asymptotic analysis.
>
> **(Regularity assumptions and convergence rate)** We note that **all** the works that provide the convergence rate adopt some regularity assumptions. These assumptions are adopted to provide the concentration analysis.
>
> We also note that there is a line of works that analyze the empirical properties of ICL [8,9]. Our work provides the **theoretical justifications** for some of the conclusions in them. For example, our result in Proposition G.5 indicates that LLMs are able to implement efficient ICL even when the examples in the prompt have _controlable_ errors, where the examples in the prompt still provide enough information to identify the desired task (Assumption G.3). This justifies that LLMs are able to implement ICL when the labels in the examples are randomly chosen from a given set [8].
>
> ___
>
> **(Novelty of Proposition 3.3)**
>
> We would like to explain the result of Proposition 3.3 here. We note that the proof of this proposition mainly uses the Bayesian rule. However, the main message of this proposition is that if the model is trained to predict the next token, and the data satisfies the latent variable model, then the perfectly pretrained LLMs will **implement BMA** for ICL. This result also serves as a **firm base** for the analysis of the attention module (Proposition 3.6) and the ICL analysis of the pretrained LLMs (Theorem 5.2). In addition, it provides a **theoretical support** for the practice of unsupervised pretraining via predicting the next token, since we show that the LLMs learn to extract the hidden concept from the prompt in this training process.
>
> ___
>
> **(Proposition 3.4 for continuous $z$)**
>
> We would like to highlight that our result in Proposition 3.4 can be **generalized** to the case where $z$ is a continuous variable under some assumptions. In fact, we need the following assumptions.
>
> **(Assumption 1)** The set $\mathfrak{Z}$ is compact and we can define norm $\|\cdot\|$ on it.
>
> This assumption states that we can properly measure the distance between the elements in $\mathfrak{Z}$. This notion of distance is used to construct the cover of $\mathfrak{Z}$.

---

> > ### Author Response · Authors · 2024-08-12
> > **Rebuttal to Reviewer He72 Part 2/2**
> >
> > **(Assumption 2)** The distribution $\mathbb{P} _{\mathcal{Z}}$ of $z$ has density function $f _{\mathcal{Z}}$ with respect to the Lebesgue measure $\mathcal{L}(\cdot)$ on $\mathcal{Z}$.
> >
> > This assumption allows us to quantify the relative magnitude of the distribution on $\mathfrak{Z}$ via its density function.
> >
> > **(Assumption 3)** The nominal distribution and the prior of $z$ are Liphschitze continuous in $z$, i.e., there exist constants $L_{1},L_{2}>0$ such that
> > \begin{align*}
> > 	\big|\log \mathbb{P}(r|z,S,c)-\log\mathbb{P}(r|z^{\prime},S,c)\big|&\leq L_{1}\|z-z^{\prime}\|, \quad
> > \big|\log f_{\mathcal{Z}}(z)-\log f_{\mathcal{Z}}(z^{\prime})\big|\leq L_{2}\|z-z^{\prime}\|
> > \end{align*}
> > for any $z,z^{\prime}\in\mathfrak{Z}$, covariate $c\in\mathfrak{X}$, response $r\in\mathfrak{X}$, and examples $S\in\mathfrak{Z}^{\ast}$.
> >
> > This assumption states the continuity of the conditional nominal distribution and the prior on $\mathfrak{Z}$. This continuity allows us to discretize $\mathfrak{Z}$ with small approximation error with respect to the distribution. We define the ball $B(z,\delta)=\{z^{\prime}\in \mathfrak{Z} | \|z-z^{\prime}\|\leq \delta \}$.
> >
> > **Proposition** Under assumptions in Proposition 3.3 and Assumptions 1 to 3, we have that
> > \begin{align*}
> >     -\sum_{t= 0}^T \log \mathbb{P}(r_{t+1} | c_{t+1}, S_t) \leq \inf_{z\in\mathfrak{Z},\delta>0 \text{ such that }B(z,\delta)\subseteq \mathfrak{Z}} \biggl[ -  \sum_{i=1}^{T +1} \log \mathbb{P}(r_i | z, S_{i-1}, c_i)\biggr] +(T+1)L_{1}\delta+L_{2}\delta+\log\frac{\mathcal{L}(\mathfrak{Z})}{\mathcal{L}(B(z,\delta))\cdot f_{\mathcal{Z}}(z)},
> >     \end{align*}
> >     where $\mathcal{L}(\cdot)$ is the Lebesgue measure on $\mathcal{Z}$.
> >
> > The proof of this result is provided in the revised manuscript.
> >
> > ___
> >
> > **(Effectiveness of experiments)**
> >
> > We would like to explain the logic behind the experiments. We note that what the reviewer says is true. However, we cannot use observations to validate a theory, we can only **refute** the theory when observing inconsistent facts. Here our experimental results **align well** with the implications of our theoretical results, which indicates that our theory is plausible. We would like to highlight that our key contribution is to provide a theory to explain ICL, which explains the observations in the experiments.
> >
> > ___
> >
> > **(Methods to improve LLM's ICL ability)**
> >
> > We would like to discuss several potential ways to improve the LLMs' ICL ability.
> >
> > First, we could further steer the subsequent output of LLMs with the **constructed hidden variables**. For example, our experiments indicate that the constructed hidden variables can make LLMs have similar ICL performance with LLMs prompted with several examples. This can be **combined with prompt engineering** to further boost the ICL ability of LLMs. This is partially explored by existing works, including [10,11].
> >
> > Second, we can utilize the constructed hidden variables to improve the efficiency of LLM agents. The LLM agents usually require **carefully designed** prompts to achieve efficient task-solving ability. Since the hidden variables indicate the task we would like the LLMs to complete, we could extract these hidden variables to **relieve the difficulty** of prompt design for agents.
> >
> > ___
> >
> > **Reference**
> >
> > [1] Xie S M, et al. An explanation of in-context learning as implicit Bayesian inference[J]. arXiv preprint arXiv:2111.02080, 2021.
> >
> > [2] Wies N, Levine Y, Shashua A. The learnability of in-context learning[J]. Advances in Neural Information Processing Systems, 2024, 36.
> >
> > [3] Wang X, Zhu W, Saxon M, et al. Large language models are implicitly topic models: Explaining and finding good demonstrations for in-context learning[C]//Workshop on Efficient Systems for Foundation Models@ ICML2023. 2023.
> >
> > [4] Jiang H. A latent space theory for emergent abilities in large language models[J]. arXiv preprint arXiv:2304.09960, 2023.
> >
> > [5] Andrews M, Vigliocco G. The hidden Markov topic model: A probabilistic model of semantic representation[J]. Topics in Cognitive Science, 2010, 2(1): 101-113.
> >
> > [6] Rahman M M, Wang H. Hidden topic sentiment model[C]//Proceedings of the 25th international conference on world wide web. 2016: 155-165.
> >
> > [7] Poria S, Cambria E, Gelbukh A, et al. Sentiment data flow analysis by means of dynamic linguistic patterns[J]. IEEE Computational Intelligence Magazine, 2015, 10(4): 26-36.
> >
> > [8] Min S, Lyu X, Holtzman A, et al. Rethinking the role of demonstrations: What makes in-context learning work?. arXiv preprint arXiv:2202.12837, 2022.
> >
> > [9] Chan, S, et al. Data distributional properties drive emergent in-context learning in transformers. Advances in Neural Information Processing Systems 35 (2022): 18878-18891.
> >
> > [10] Liu S, Xing L, Zou J. In-context vectors: Making in-context learning more effective and controllable through latent space steering[J]. arXiv preprint arXiv:2311.06668, 2023.
> >
> > [11] Todd E, et al. Function vectors in large language models. arXiv preprint arXiv:2310.15213, 2023.

---

### Decision · Action_Editor_Guwd · 2024-09-11

**Recommendation:** Reject

**Comment:**

All three reviewers provided a recommendation of "leaning accept" in their final recommendations, praising the interesting research questions considered, the theoretical treatment of a problem that has mostly previously only been explored empirically, and some interesting accompanying experimental results.  Concerns though were raised about the strength of the assumptions made by the theory, the discussion of related work, the clarity of the presentation, and the real world relevance of some of the results.

However, having read through the paper myself, I, unfortunately, have some major concerns of my own, which closely relate to some of those of the reviewers but which I see myself as being more serious than they may have done. I also feel like some critical issues that were raised were not properly addressed in the rebuttal/revision.  Due to these concerns, which are explained below, I am afraid that I do not think the paper is currently suitable for publication at TMLR.  There are though certainly significant positive aspects of the work and I strongly encourage the authors to resubmit with major revisions, the changes needed are simply too large to deal with in "minor revisions".

In short, I think the paper has three critical flaws at the moment:
1. It fails to properly discuss large quantities of related work in sufficient detail.  In particular, there is work that gives strong evidence against many of the claims being made.  Perhaps most notable here is [Kossen et al 2024] (which is not cited at all), but it is far from the only one, with the Min et al 2022 paper included in the references but never discussed another clear example.  I see the inclusion of a discussion of these other viewpoints as an essential change that would be required for the paper to be published.  Similar issues were raised by the reviews, but I do not feel that the revisions done so far are close to being sufficient to correct the issue: a full section discussing the related work and other arguments that have been put forward (in particular alternative hypotheses to BMA) is needed.
2. I do not feel the assumptions made by the theory are reasonable and I think this leaves some of the core results somewhat vacuous, and certainly not sufficient to justify the strong arguments the authors aim to make.  In particular, I think that Assumption 3.1 essentially equates to already assuming that the ICL is a BMA, such that Proposition 3.3 is a circular argument rather than evidence that ICL is performing BMA, with the somewhat trivial proof reflecting this. Moreover, I am not convinced that Assumption 3.1 is reasonable for true LLMs.  An LLM has the ability to switch between different concepts.  For example, if you suddenly start feeding it prompts for a new problem or start giving it inverted responses.  The given setup assumes that the previous responses only affect the learning through z_* and some fixed dynamics of a hidden state that is not directly a function of the context, but the LLM cannot be making this assumption as it needs to be keeping open the possibility that z_* actually changes itself during the ICL.  In particular, I think the results of Kossen et al 2024 provide strong empirical evidence that Assumption 1 is not true for an LLM by showing how it is capable of switching its "concept" during ICL (e.g. switching between giving the correct answer and the opposite of the correct answer during ICL).  Thus, I think the shortfall in the assumptions is much deeper than the obvious fact that pretraining won't be perfect (Assumption 3.2).
3. The core claims made by the paper are far too strong and not justified as per point 2 above.  Namely, I do not think the paper can be published while making the claim "LLMs implement BMA for ICL".  This would be a profound assertion if true, but I do not believe it is and in my opinion, this means the paper very clearly fails TMLR's criteria on claims and evidence at present.  For example, [Kossen et al 2024] essentially already empirically falsified this claim by showing that information is not treated equally during learning and ICL does not seem able to fully overcome the original prior preferences (and certainly does not follow the suggested regret convergence).  There is potentially some room for debate here about the claim and I am not saying that the paper needs to agree with their viewpoint, but the presentation of this claim as simply being fact is not acceptable if the paper is to be published.  Again, I feel similar issues were raised by reviewers but have not been sufficiently addressed in the revision.

While the second flaw is not necessarily fatal as there are other contributions in the paper as well (and the assumptions being made here are still a weakening of assumptions from previous work, even if they are potentially circular), I do not think the paper can be accepted at TMLR with 1. and 3. still outstanding.  Correcting these, and adding more discussions about the limitations of the assumptions to give context relating to issue 2, are thus the key revisions I would like to see for a revised version of the paper to be accepted.

If the authors choose to revise and resubmit the work, I would also recommend working on improving the general writing of the paper as well.  Large parts of the paper are quite difficult to follow and overly dense (e.g. I found the abstract difficult to parse) and there are lots of lower-level grammatical errors as well.  I also feel that more could be done to address some of concerns by the reviewers as well.  As a more minor note, I would suggest including the appendices in the main submission file for ease of reading.

[Kossen et al.  In-Context Learning Learns Label Relationships but is Not Conventional Learning. ICLR 2024]

**Audience:**

I believe the paper will be of interest to the TMLR audience: it is on a hot topic at the moment and there are new results that will be of interest.

**Claims And Evidence:**

I do not believe the paper meets the TMLR criteria on claims and evidence, please see the comments below.

**Resubmission Of Major Revision:**

The authors may consider submitting a major revision at a later time.